# Heterogeneity in surface sensing suggests a division of labor in *Pseudomonas aeruginosa* populations

Catherine R Armbruster[1†], Calvin K Lee[2,3,4†], Jessica Parker-Gilham[1], Jaime de Anda[2,3,4], Aiguo Xia[5], Kun Zhao[6,7], Keiji Murakami[8], Boo Shan Tseng[9], Lucas R Hoffman[1,10], Fan Jin[5,11]*, Caroline S Harwood[1], Gerard CL Wong[2,3,4]*, Matthew R Parsek[1,12]*

[1]Department of Microbiology, University of Washington, Seattle, United States; [2]Department of Bioengineering, University of California, Los Angeles, Los Angeles, United States; [3]Department of Chemistry and Biochemistry, University of California, Los Angeles, Los Angeles, United States; [4]California NanoSystems Institute, University of California, Los Angeles, Los Angeles, United States; [5]Hefei National Laboratory for Physical Sciences at the Microscale, University of Science and Technology of China, Hefei, China; [6]Key Laboratory of Systems Bioengineering (Ministry of Education), School of Chemical Engineering and Technology, Tianjin University, Tianjin, China; [7]Collaborative Innovation Centre of Chemical Science and Engineering, Tianjin University, Tianjin, China; [8]Department of Oral Microbiology, Institute of Biomedical Sciences, Tokushima University Graduate School, Tokushima, Japan; [9]School of Life Sciences, University of Nevada, Las Vegas, United States; [10]Department of Pediatrics, University of Washington, Seattle, United States; [11]Institute of Synthetic Biology, Shenzhen Institutes of Advanced Technology, Chinese Academy of Sciences, Shenzhen, China; [12]Integrative Microbiology Research Centre, South China Agricultural University, Guangzhou, China

*For correspondence:
fjinustc@ustc.edu.cn (FJ);
gclwong@seas.ucla.edu (GCLW);
parsem@uw.edu (MRP)

†These authors contributed equally to this work

Competing interests: The authors declare that no competing interests exist.

**Abstract** The second messenger signaling molecule cyclic diguanylate monophosphate (c-di-GMP) drives the transition between planktonic and biofilm growth in many bacterial species. *Pseudomonas aeruginosa* has two surface sensing systems that produce c-di-GMP in response to surface adherence. Current thinking in the field is that once cells attach to a surface, they uniformly respond by producing c-di-GMP. Here, we describe how the Wsp system generates heterogeneity in surface sensing, resulting in two physiologically distinct subpopulations of cells. One subpopulation has elevated c-di-GMP and produces biofilm matrix, serving as the founders of initial microcolonies. The other subpopulation has low c-di-GMP and engages in surface motility, allowing for exploration of the surface. We also show that this heterogeneity strongly correlates to surface behavior for descendent cells. Together, our results suggest that after surface attachment, *P. aeruginosa* engages in a division of labor that persists across generations, accelerating early biofilm formation and surface exploration.

## Introduction

*Pseudomonas aeruginosa* is an opportunistic pathogen that engages in a range of surface-associated behaviors and is a model bacterium for studies of surface-associated communities called biofilms. Biofilms are dense aggregates of cells producing extracellular matrix components that hold the community together. The biofilm mode of growth is beneficial for bacteria in that it allows cells to

**eLife digest** Bacteria can adopt different lifestyles, depending on the environment in which they grow. They can exist as single cells that are free to explore their environment or group together to form 'biofilms'. The bacteria in biofilms stick to a surface, and produce a slimy 'matrix' that covers and thereby protects them. Biofilms have been found in lung infections that affect people with the genetic disorder cystic fibrosis, and can also form on the surface of medical implants. Because the biofilm lifestyle protects bacteria from the immune system and antimicrobial drugs, learning about how biofilms form could help researchers to discover ways to prevent and treat such infections.

Many bacteria switch between the free-living and biofilm lifestyles by altering their levels of a signaling molecule called cyclic diguanylate monophosphate (called c-di-GMP for short). Bacteria living in biofilms have much higher levels of c-di-GMP than their free-living counterparts, and bacteria that have high levels of c-di-GMP produce more biofilm matrix.

Bacteria called *Pseudomonas aeruginosa* use a protein signaling complex called the Wsp system to sense that they are on a surface and increase c-di-GMP production. Questions remained about how quickly this change in production occurs, and whether bacteria pass on their c-di-GMP levels to the new descendant cells when they divide.

Armbruster et al. monitored individual cells of *P. aeruginosa* producing c-di-GMP as they began to form biofilms. Unexpectedly, not all cells increased their c-di-GMP levels when they first attached to a surface. Instead, Armbruster et al. found that there are two populations – high and low c-di-GMP cells – that each perform complementary and important tasks in the early stages of biofilm formation.

The high c-di-GMP cells represent 'biofilm founders' that start to produce the biofilm matrix, whereas the low c-di-GMP cells represent 'surface explorers' that spend more time traveling along the surface. Armbruster et al. found that the Wsp surface sensing system generates these two populations of cells. Moreover, the c-di-GMP levels in a bacterial cell even affect the behavior of the descendant cells that form when it divides. This effect can persist for several cell generations.

More work is needed to examine exactly how the biofilm founders and surface explorers interact and influence how biofilms form, and to discover if blocking c-di-GMP signaling prevents biofilm formation. This could ultimately lead to new strategies to prevent and treat infections in humans.

maintain close proximity to nutrients, promotes exchange of genetic material, and confers cells protection from a variety of chemical and environmental stresses (e.g. nutrient limitation, desiccation, and shear forces), as well as engulfment by protozoa in the environment or by phagocytes in a host (*Davey and O'toole, 2000*). Collectively, these advantages make biofilm formation integral to prokaryotic life.

The secondary messenger signaling molecule cylic diguanylate monophosphate (c-di-GMP) controls the transition between the planktonic to the biofilm mode of growth. In many bacterial species, including *P. aeruginosa*, elevated c-di-GMP results in repression of flagellar motility genes, while promoting expression of genes involved in producing a biofilm matrix (*Römling et al., 2013*). The *P. aeruginosa* biofilm matrix is composed of a combination of polysaccharides (including Pel and Psl), proteins (including the adhesin CdrA), and extracellular DNA (*Ma et al., 2009*). Biofilm matrix production is an energetically costly process that is regulated at multiple levels (*Wei and Ma, 2013*). The *cdrA*, *pel* and *psl* genes are all transcriptionally induced under conditions of high c-di-GMP (*Starkey et al., 2009*).

For many species, the initial step in biofilm formation involves adherence of free swimming planktonic cells to a surface and the initiation of surface sensing. *P. aeruginosa* has at least two distinct surface sensing systems, the Wsp and the Pil-Chp systems, that when activated, lead to biofilm formation. The Wsp system senses an unknown surface-related signal (recently proposed to be membrane perturbation [*Chen et al., 2014*]) through WspA, a membrane-bound protein homologous to methyl-accepting chemotaxis proteins (MCPs). Activation of this system stimulates phosphorylation of the diguanylate cyclase WspR, which leads to the formation of aggregates of phosphorylated WspR (WspR-P) in the form of visible subcellular clusters. This aggregation of WspR-P potentiates its activity, increasing c-di-GMP synthesis (*Huangyutitham et al., 2013*). In comparison, the Pil-Chp

chemosensory-like system initiates a hierarchical cascade of second messenger signaling in response to a surface (*Luo et al., 2015*). First, an increase in cellular cAMP levels occurs through activation of the adenylate cyclase CyaB by the Pil-Chp complex. This increases expression of genes involved in type IV pilus biogenesis, including PilY1. PilY1 is associated with the type IV pilus and harbors a Von Willebrand motif, which is involved in mechanosensing in eukaryotic systems (*Kuchma et al., 2010*). Thus, it has been proposed that this protein may be involved in the mechanosensing of surfaces (*Persat et al., 2015*). The output of this second signal is through the diguanylate cyclase, SadC, resulting in an increase in cellular c-di-GMP levels. Unlike the Wsp system, which localizes laterally along the cell (*O'Connor et al., 2012*), PilY1 is required to be associated with polarly-localized type IV pili in order to stimulate c-di-GMP production (*Luo et al., 2015*; *Kuchma et al., 2010*), suggesting that *P. aeruginosa* deploys both polar and laterally localized systems to promote c-di-GMP synthesis in response to a surface.

Bacteria in biofilms have long been appreciated to exhibit phenotypic heterogeneity due to chemical variation within the biofilm itself, including gradients of oxygen (*Wessel et al., 2014*), nutrients (*Schreiber et al., 2016*), and pH (*Vroom et al., 1999*). These environmental conditions are sensed by individual bacterial cells, leading to differential gene expression and metabolic activities even within a genetically homogeneous population (*Stewart and Franklin, 2008*). Specifically, the term 'division of labor' refers to cases where genetic or phenotypic heterogeneity results in subpopulations of cells cooperating to perform distinct tasks that provide an overall fitness benefit to the population (*West and Cooper, 2016*). Through task allocation, subpopulations of cells can engage in behaviors that are impossible to perform simultaneously (e.g. bet-hedging strategies between biofilm and planktonic cells [*Lowery et al., 2017*]), energetically costly to switch between (e.g. task-switching [*Goldsby et al., 2012*]), or are metabolically incompatible (e.g. photosynthesis and nitrogen fixation in cyanobacteria ([*Rossetti et al., 2010*]). In particular, there is a rich body of literature demonstrating that genetic (*Kim et al., 2016*; *Dragoš et al., 2018*; *Klausen et al., 2003*) and phenotypic variation (*Klauck et al., 2018*; *Serra et al., 2015*; *Haagensen et al., 2007*) in surface motility and polysaccharide production among individual bacterial cells within a biofilm can represent a division of labor that is required to achieve the architecture and structural integrity of the biofilm matrix (*van Gestel et al., 2015*).

Beyond heterogeneity as a result of variation in environmental signals, recent single-cell analyses have revealed that c-di-GMP signaling can drive phenotypic heterogeneity among populations of single cells exposed to the same environmental inputs. Planktonic *P. aeruginosa* have been shown to achieve heterogeneity among very low levels of c-di-GMP through assymetrical partitioning of a diguanylate cyclase during cell division, leading to diverse swimming molitity behaviors (*Kulasekara et al., 2013*). More recently, this same assymetrical cell division mechanism was shown to generate two populations of *P. aeruginosa*, one piliated and one flagellated, that are each required for efficient tissue colonization (*Laventie et al., 2019*). Together, these studies support a role for c-di-GMP heterogeneity in generating diverse bacterial behaviors during both biofilm and planktonic growth.

Here, we examined the dynamics of c-di-GMP production and bacterial surface motility at the single-cell level during early stages of biofilm formation. We used a plasmid-based, transcriptional reporter of intracellular c-di-GMP to follow the downstream fate of cells producing varying levels of c-di-GMP in response to surface attachment. Within a clonal population of *P. aeruginosa*, we found that levels of c-di-GMP vary among individual cells as they sense a surface, leading to a division of labor between two energetically costly behaviors associated with early biofilm formation: surface exploration and polysaccharide production.

## Results

### Cellular c-di-GMP levels rapidly increase upon surface attachment

We initially compared levels of c-di-GMP between *P. aeruginosa* PAO1 cells growing attached to a silicone surface and subjected to constant flow for 4 hr to those grown planktonically for 4 hr. As expected, we observed that PAO1 cellular c-di-GMP levels are 4.4-fold higher ($\pm$0.78 SD, N = 3, p$\leq$0.05) after 4 hr of growth attached to a surface compared to planktonic growth (*Figure 1A*). Because direct measurement of c-di-GMP by LC-MS/MS is limited by our ability to generate enough

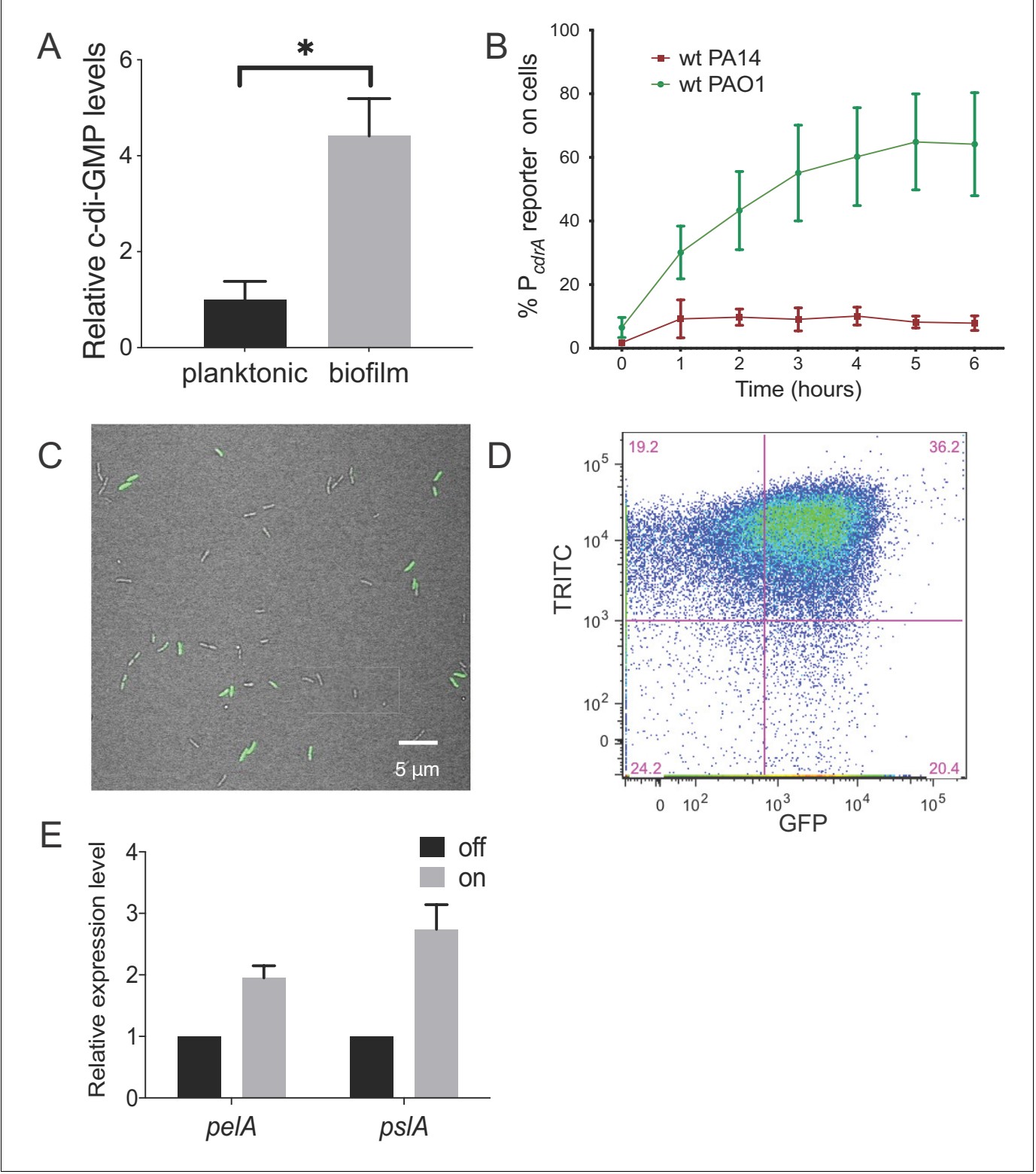

**Figure 1.** Heterogeneity in cellular levels of c-di-GMP during early *P. aeruginosa* biofilm formation. (**A**) c-di-GMP levels are elevated rapidly upon association of *P. aeruginosa* PAO1 cells with a surface. Relative levels of intracellular c-di-GMP in wild type PAO1 cells grown either planktonically or after 4 hr of attachment to a silicone tube. Values are normalized to the average concentration of c-di-GMP in planktonic cells, in pmol c-di-GMP/mg total protein as determined by LC-MS/MS, and presented as mean and SD. *p<0.05 by T-test, N = 3. *Figure 1—figure supplement 1* shows the Pel polysaccharide operon is transcriptionally activated almost 10-fold compared to planktonic cells within 30 min of surface attachment. (**B**) Two commonly

*Figure 1 continued on next page*

*Figure 1 continued*

studied *P. aeruginosa* lab strains, PAO1 and PA14, differentially activate the c-di-GMP reporter during surface sensing. Wild type PAO1 or PA14 cells harboring the c-di-GMP reporter (pP*cdrA*::*gfp*ASV) were grown to mid-log phase in planktonic culture, then inoculated into a flow cell and supplied with 1% LB medium. Surface attached cells were imaged immediately after inoculation (time 0 hr), and hourly for 12 hr. The c-di-GMP reporter is activated in a subset of wild type PAO1 cells within 1 hr of surface attachment and remains activated in approximately 60% of PAO1 cells during the first 6 hr of attachment. In PA14, the c-di-GMP reporter is activated in a smaller proportion of attached cells compared to PAO1. Data points are mean percentage of reporter activated cells from each time point across at least three biological replicates, with standard deviation. *Figure 1—figure supplement 4* shows an additional c-di-GMP responsive transcriptional reporter (using the *siaA* promoter) is also responsive to Wsp-dependent changes in cellular levels of c-di-GMP. (C) Wild type PAO1 cells display heterogeneity in c-di-GMP reporter activity after 6 hr of surface attachment. Confocal microscopy image of wild type PAO1 P*cdrA*::*gfp*ASV grown in 1% LB after 6 hr of surface attachment during a time course flow cell experiment. Bright field (gray) and GFP (green) channels are merged. Wild type PAO1 P*cdrA*::*gfp*ASV was grown in 1% LB and imaged by CSLM. *Figure 1—figure supplement 2* shows additional representative timecourse images of PAO1. (D) Psl exopolysaccharide production is enriched in the population of cells with high c-di-GMP. Representative scatterplot of reporter activity versus Psl lectin binding in wild type PAO1 harboring the pP*cdrA*::*gfp*ASV reporter grown for four hours in LB before surface attached cells were harvested, stained with the lectin, washed, and counted by flow cytometry. (E) Subpopulations of PAO1 cells with high and low c-di-GMP reporter activity are physiologically distinct. Cells with higher c-di-GMP reporter activity have increased expression of Pel and Psl biosynthetic machinery genes. After 4 hr of attachment to glass, wild type PAO1 cells were separated by flow-assisted cell sorting (FACS) into a population of cells with high (on) and low (off) c-di-GMP reporter activity, then qRT-PCR was performed to quantify expression of Pel and Psl exopolysaccharide biosynthesis genes. Levels of expression of Pel or Psl mRNA were normalized to the off population. *p<0.05 by T-test, N = 3 biological replicates. *Figure 1—figure supplement 3* shows controls for validating the protocol to monitor pP*cdrA*::*gfp*ASV by flow cytometry. *Figure 1—source data 2* shows by flow cytometry that Psl and Pel polysaccharide production is highest in cells with high pP*cdrA*::*gfp*ASV reporter activity.

The online version of this article includes the following source data and figure supplement(s) for figure 1:

**Source data 1.** Raw data for fluorescence intensities.
**Source data 2.** Distributions of median fluorescence intensity values of PAO1 WT pP*cdrA*::*gfp*ASV cells obtained via bootstrap sampling.
**Figure supplement 1.** The c-di-GMP-regulated promoter of the Pel polysaccharide operon is transcriptionally activated almost 10-fold compared to planktonic cells within 30 min of attachment of PAO1 to a silicone tube.
**Figure supplement 2.** Representative time course images showing the c-di-GMP reporter (P*cdrA*::*gfp*ASV) transitioning from inactive upon initial attachment of wild type PAO1 (0 hr) to active in a subpopulation of cells between 1 and 12 hr during a flow cell experiment.
**Figure supplement 3.** Fluorescence intensities of individual wild type PAO1 pP*cdrA*::*gfp*ASV cells imaged over 12 hr following surface attachment in a flow cell experiment (n ≥ 3 experiments per time point).
**Figure supplement 4.** P-values for Kruskal-Wallis test for comparing medians observed in *Figure 1—figure supplement 3*, followed by Tukey's post-test for multiple comparisons.
**Figure supplement 5.** Development of a protocol to monitor pP*cdrA*::*gfp*ASV using an LSRII flow cytometer.
**Figure supplement 6.** The *siaA* promoter, regulated by c-di-GMP binding to FleQ, is responsive to Wsp-dependent changes in cellular levels of c-di-GMP.
**Figure supplement 7.** Psl and Pel polysaccharide production is highest in cells with high c-di-GMP as measured by the pP*cdrA*::*gfp*ASV reporter.
**Figure supplement 8.** Development of a protocol to sort biofilm cells by pP*cdrA*::*gfp*ASV using flow assisted cell sorting (FACS).
**Figure supplement 9.** Flow cytometry control depicting the specificity of TRITC-HHA lectin for Psl.

biomass at earlier time points, we used qRT-PCR to monitor *pel* transcript levels as a readout of c-di-GMP. We found that after just 30 min of surface attachment, *pelA* transcript levels had increased almost 10-fold compared to planktonically grown cells (*Figure 1—figure supplement 1*). This is consistent with previously published literature showing that transcription of the *pel* operon is directly and positively controlled by high cellular levels of c-di-GMP (*Hickman and Harwood, 2008*; *Baraquet et al., 2012*).

## The P*cdrA*::*gfp* reporter suggests heterogeneity in c-di-GMP levels during surface sensing

Next, we sought to visualize early c-di-GMP signaling events at the single cell level. To this end we used a plasmid-based, c-di-GMP responsive transcriptional reporter, pP*cdrA*::*gfp*ASV (*Rybtke et al., 2012*) in two commonly-studied *P. aeruginosa* strains, PAO1 and PA14. Planktonic cells (a condition where the reporter is inactive due to low c-di-GMP levels) were used to inoculate flow cell chambers. We imaged individual cells of each reporter strain hourly for up to 6 hr after surface attachment (*Figure 1B* and *Figure 1—figure supplement 2*). As expected, we saw minimal GFP fluorescence at the 0 hr time point (right after surface attachment). However, by 1 hr, the reporter was activated in a subset of surface attached cells, as defined by background subtracted GFP fluorescence ≥321 fluorescence units (referred to as reporter 'on' subpopulations). Interestingly, between 4 and 6 hr post

inoculation, we consistently observed that the c-di-GMP reporter was only active in a subset of cells in both strains (*Figure 1C*). In PA14, the reporter was activated in 10% of the population over 6 hr, whereas PAO1 displayed greater reporter activity, with 40–60% of the cells displaying reporter activity through 12 hr (*Figure 1B*). An analysis of single cell fluorescence supported these observations. We plotted each cell's individual fluorescence values over time and observed populations of cells with low and high c-di-GMP reporter activity at each timepoint after the 0 hr (*Figure 1—figure supplement 3*). We also observed long tails of high GFP fluorescence, particularly at 2 and 4 hr. This suggests that the reporter 'on' subpopulation represents cells with a range of high c-di-GMP levels. This wide range of high reporter activity cells between 2 and 4 hr could be indicative of an early 'spike' in c-di-GMP production that levels off over time (*Figure 1—figure supplement 3*).

We next examined the change in distribution of fluorescence intensity over time for single cells by bootstrap sampling of the single cell fluorescence intensity values (*Figure 1—figure supplement 4*). The purpose of this bootstrapping analysis is to examine whether the distributions of fluorescence intensity differ at each time point. We found that the median fluorescence intensity was significantly different at every time point except between 4 and 6 hr, and between 8, 10, and 12 hr (*Figure 1—source data 2*). Together, these single cell analyses support a model in which c-di-GMP signaling is initiated rapidly upon surface attachment in the first 4–6 hr for a subpopulation of attached cells, while the rate of c-di-GMP increase tends to level off at later timepoints.

We confirmed the microscopy results from comparing PAO1 and PA14 reporter fluorescence using flow cytometry to assess the proportion of attached cells that were fluorescent (*Figure 1—figure supplement 5D, E*). To be sure that the promoter of *cdrA* is representative of c-di-GMP-regulated gene expression, we replaced P*cdrA* with the promoter of *siaA*, a gene that is also highly expressed under conditions of elevated c-di-GMP (*Starkey et al., 2009*; *Baraquet and Harwood, 2016*). We found that pP*siaA*::*gfp*ASV reporter activity resembled that of pP*cdrA*::*gfp*ASV in response to a surface (*Figure 1—figure supplement 6*). Thus, reporter activity is indeed linked to cellular levels of c-di-GMP.

## Cyclic di-GMP heterogeneity leads to phenotypic diversification at early stages of biofilm formation

We then wanted to confirm that subpopulations of surface-attached *P. aeruginosa* cells with high and low c-di-GMP reporter activity are truly physiologically distinct from one another. We used TRITC-labeled lectins to stain for two c-di-GMP-induced exopolysaccharides, Psl and Pel (*Zhao et al., 2013*; *Jennings et al., 2015*), the presence of which is indicative of biofilm formation by PAO1 and PA14, respectively. After 4 hr of attachment to glass, we observed an enrichment of TRITC-conjugated lectin staining in the population of cells with high c-di-GMP reporter activity (*Figure 1D* and *Figure 1—figure supplement 7*), demonstrating that the subpopulation of cells with high c-di-GMP is producing more exopolysaccharide than their low c-di-GMP counterparts. This correlation was weaker for Psl than Pel, probably due to the fact planktonic populations can make low levels of Psl (though not the case for Pel) (*Wei and Ma, 2013*). As a complementary approach, we separated 4 hr surface-grown cells of the reporter strain into reporter 'on' and 'off' subpopulations using flow-assisted cell sorting (FACS; *Figure 1—figure supplements 8* and *9*). We then applied qRT-PCR to compare Pel and Psl transcript levels in these two populations. Both the *pel* and *psl* operon transcripts were elevated in the reporter 'on' subpopulation, relative to the reporter 'off' subpopulation (*Figure 1E*). These data support that, with respect to c-di-GMP signaling, there are at least two distinct subpopulations that arise shortly after surface attachment.

## The Wsp system is required for surface sensing

We next evaluated the relative contributions of the Wsp and Pil-Chp surface sensing systems to surface-induced c-di-GMP production. Strains with mutations in the Pil-Chp chemosensory system were not significantly defective in surface sensing activity. Deletion of the diguanylate cyclase activated through the Pil-Chp system (PAO1 Δ*sadC*) and the gene encoding the putative sensor PilY1 (PAO1 Δ*pilY1*) did not significantly influence reporter activity in response to a surface (*Figure 2—figure supplement 1A,B*). Whereas both the SadC and PilY1 mutants displayed wild type levels of reporter activity, a mutant lacking the main Type IV pilus filament protein (PAO1 Δ*pilA*) did show a statistically significant defect in reporter activity by 6 hr (*Figure 2—figure supplement 1B*; p<0.05 by T-test).

We then mutated the c-di-GMP cyclase gene, *wspR,* to inactivate the Wsp system. In addition, we deleted the gene encoding the methylesterase *wspF*, which locks the system into the active state, regardless of whether cells are surface-associated. We found that the PAO1 Δ*wspR* strain exhibited extremely low levels of reporter activity during the first 6 hr after surface attachment (*Figure 2A* and *Figure 2—figure supplement 2*). Complementation of PAO1 Δ*wspR* restored wild type levels of activity at all time points (*Figure 2—figure supplement 3*). As expected, PAO1 Δ*wspF* had a high proportion of reporter active cells (*Figure 2A*). We repeated these experiments in the lab strain PA14 and saw a similar trend for Wsp mutants (*Figure 2—figure supplement 4*). Kulesekara *et al.* (*Kulasekara et al., 2013*) used a FRET-based c-di-GMP reporter to show that planktonic *P. aeruginosa* has heterogeneous, albeit very low, concentrations of c-di-GMP which it achieves through assymetrical partitioning of a phosphodiesterase (PA5017, also called Pch or DipA) to the flagellated cell pole during cell division. However, we found no evidence of heterogeneity in c-di-GMP within the planktonic cell inoculum contributing to our observations. We ruled out the phosphodiesterase (PA5017) as responsible for the heterogeneity we see during surface sensing by examining a deletion mutant of PA5017 and showing that this strain still exhibited heterogeneity during surface sensing (*Figure 2—figure supplement 5*).

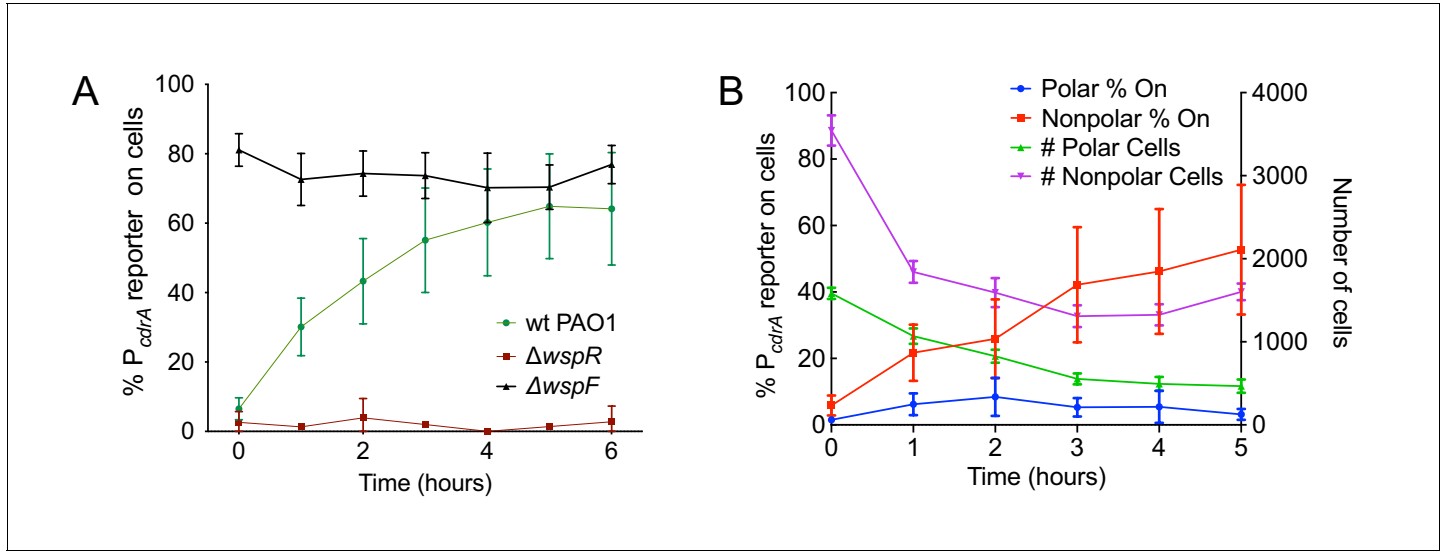

**Figure 2.** The Wsp system generates heterogeneity in cellular levels of c-di-GMP during early *P. aeruginosa* biofilm formation. (**A**) The Wsp system is required for activation of the pP*cdrA*::*gfp*ASV reporter during surface sensing. Six hour time course plot of the average percentage of surface-attached cells from either wild type PAO1 (green), PAO1 Δ*wspR* (red), or PAO1 Δ*wspF* (black) in which the pP*cdrA*::*gfp*ASV reporter had turned 'on' at each hour. Cells were identified as 'on' if their average GFP fluorescence was greater than twice the average background GFP fluorescence of the image. Error bars = standard deviation. N ≥ 3 biological replicates. See *Figure 2—figure supplement 1* for the same timecourse using mutants in the Pil-Chp surface sensing system. *Figure 2—figure supplement 2* shows representative images from *Figure 2A*. *Figure 2—figure supplement 3* shows that complementing the *wspR* mutant restores wild type levels of reporter activity. *Figure 2—figure supplement 4* shows that the lab strain PA14 also displays Wsp-dependent c-di-GMP heterogeneity. *Figure 2—figure supplement 5* shows that *PA5017* (Pch) is not required for heterogeneity during surface sensing. (**B**) Laterally attached cells have higher c-di-GMP levels than polarly attached cells. Five hour time course plot depicting, on the left axis, the percentage of pP*cdrA*::*gfp*ASV reporter 'on' cells that were either polarly (blue) or laterally (red) attached to the surface of a glass coverslip in a flow cell at each hour. The right axis depicts the total number of polar (green) and laterally attached (purple) cells at each time point. Cells were identified as pP*cdrA*:*gfp*ASV reporter 'on' if their average GFP fluorescence was greater than 321 fluorescence units. Error bars = standard deviation. N = 4 biological replicates.

The online version of this article includes the following figure supplement(s) for figure 2:

**Figure supplement 1.** Mutants predicted to inactivate the Pil-Chp surface sensing system largely retain pP*cdrA*::*gfp*ASV reporter activity during the first six hours of surface sensing.

**Figure supplement 2.** The pP*cdrA*::*gfp*ASV reporter is sensitive to Wsp-dependent variation in c-di-GMP during surface sensing.

**Figure supplement 3.** Complemented diguanylate cyclase mutants display wild type levels of P*cdrA*::*gfp*ASV reporter activity.

**Figure supplement 4.** Activity of the pP*cdrA*::*gfp*ASV reporter in strain PA14 is dependent on the Wsp system.

**Figure supplement 5.** A mutation in PA5017 (also referred to as *pch* or *dipA*) does not impact the heterogeneity observed in surface sensing.

Since the Pil-Chp surface sensing apparatus is polarly localized and the Wsp system is localized laterally along the length of the cell body, we examined whether reporter activity correlated with polar versus lateral attachment to the surface. We found that reporter activity was very low in polarly attached cells, while cells attached along the entire length of the cell body displayed a higher proportion of reporter-activated cells at that same time point (*Figure 2B*). (Our analysis does not account for the time period each cell spends in either the polarly or non-polarly attached states.) This finding is consistent with the localization of the Wsp system and its role for early c-di-GMP signaling during surface sensing.

## Heterogeneity in c-di-GMP levels among cells correlates with Wsp system activity

The specific activity of purified WspR increases as a function of WspR concentration when the protein is treated with beryllium fluoride to mimic phosphorylation, supporting the idea that formation of subcellular clusters of WspR-P potentiates its diguanylate cyclase activity and leads to elevated c-di-GMP (*Huangyutitham et al., 2013*). Fewer than 1% of wild-type cells grown in broth have a visible WspR-YFP cluster. However, after a short period of growth on an agar surface, WspR-YFP clusters were visible in 30–40% of wild type PAO1 cells, and this is dependent on sensing by the membrane-bound protein WspA, which is laterally distributed in cells (*Kuchma et al., 2010*). To directly link WspR cluster formation with diguanylate cyclase activity at the cellular level and with surface sensing, we constructed a version of the c-di-GMP reporter that expresses mTFP1 instead of GFP (pP$_{cdrA}$::*mTFP1*) to avoid the issue of spectral overlap with WspR-YFP. We monitored reporter activity in two point mutants of WspR (L170D and E253A) that are driven by an inducible promoter, translationally fused to eYFP, and have been previously shown to form large subcellular WspR clusters in a higher percentage of cells than wild-type WspR. The WspR[L170D] protein is highly active for c-di-GMP production, and it forms subcellular clusters in about 75% of agar surface-grown cells. A WspR[E253A] point mutation abolishes diguanylate cyclase activity, but this protein still forms clusters in about 70% of surface-grown cells (*Huangyutitham et al., 2013*). As expected, in the presence of inducer, we observed a large increase in c-di-GMP reporter activity in WspR[L170D], but not WspR[E253A] (*Figure 3A,B*). We then asked whether the heterogeneity in reporter activity in response to surface attachment correlates with WspR clustering in the WspR[L170D] strain. We found that pP$_{cdrA}$::*mTFP1* activity was significantly higher in cells with at least one subcellular WspR-eYFP focus in the WspR[L170D] strain compared to cells without a WspR-eYFP focus (*Figure 3C* and *Figure 3—figure supplement 1*). These data indicate that the heterogeneity observed in c-di-GMP signaling after surface attachment is due to the heterogeneity in the activity of the Wsp system, as reflected by subcellular clustering of active WspR-P.

For phenotypic heterogeneity to represent a division of labor, it must result in a fitness benefit to the population. Therefore, we next asked whether the observed heterogeneity in c-di-GMP signaling in response to a surface has a meaningful influence on biofilm formation. This was particularly important since previously published results indicated that a *wspR* mutation had only a small impact on biofilm production (*Kulasakara et al., 2006*). However, these studies assessed biofilm formation at later stages of biofilm growth that were well beyond initial surface attachment. Therefore, we chose to compare a *wspR* mutant to wild type at earlier biofilm stages. We performed *in vitro* biofilm assays and observed that a PAO1 Δ*wspR* mutant was defective for biofilm formation relative to wild type PAO1 at 2, 4, and 6 hr post-attachment (*Figure 4A*). However, at later stages of development (24 hr), the *wspR* mutant caught up and produced similar amounts of biofilm biomass relative to wild type . Complementation of the Δ*wspR* strain *in trans* restored wild type levels of biofilm formation at all time points (*Figure 4A*). These data suggest that the Wsp system rapidly responds to surface contact to generate elevated levels of c-di-GMP in a subpopulation of cells, which accelerates biofilm production. Given the importance of c-di-GMP signaling in biofilm production, the fact that the Δ*wspR* strain can ultimately attain wild-type levels of biofilm biomass suggests that one of the many other known c-di-GMP cyclases present in *P. aeruginosa* may ultimately compensate for c-di-GMP production in the absence of WspR.

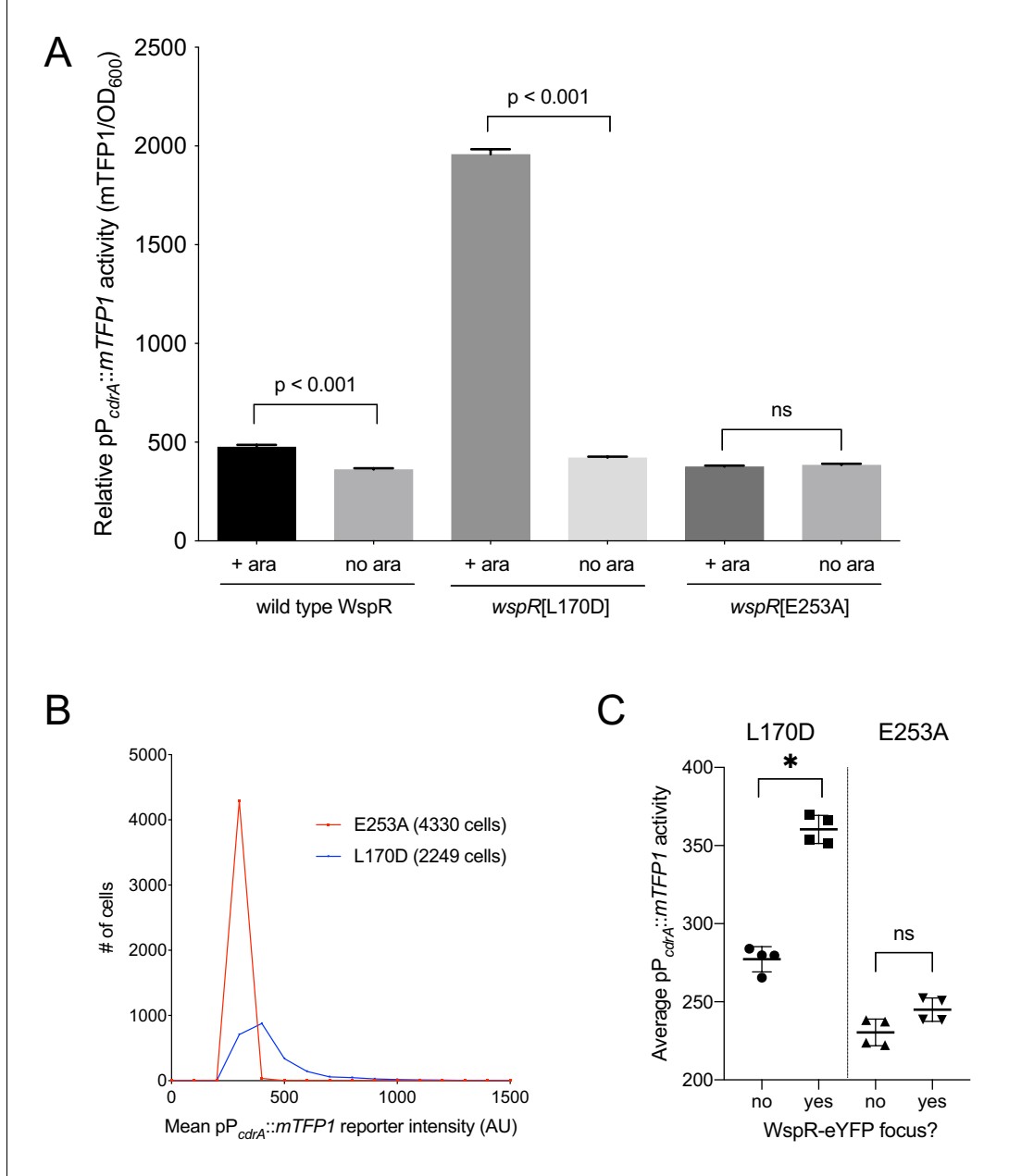

**Figure 3.** Activity of the pP$_{cdrA}$::*mTFP1* reporter is dependent on the ability of WspR to produce c-di-GMP. (**A**) The pP$_{cdrA}$::*mTFP1* reporter is active in surface grown cells with functional, arabinose-inducible alleles of WspR when arabinose is added to the media. Wild type WspR represents the strain PAO1 Δ*wspR* attCTX::*wspR*-eYFP. *wspR*[L170D] represents the strain PAO1 Δ*wspR* attCTX::*wspR*[L170D]-eYFP, which produces large subcellular clusters of WspR and grows as rugose small colonies on LB with 1% arabinose, a phenotype that is indicative of high intracellular c-di-GMP. *wspR*[E253A] represents the strain PAO1 Δ*wspR* attCTX::*wspR*[L170D]-eYFP cells, which forms large subcellular WspR clusters, but does not produce c-di-GMP via WspR due to the point mutation located in its active site. Cells were grown on LB agar plates with 100 μg/mL gentamicin, and in the presence or absence of 1% arabinose. Cells were resuspended in PBS and mTFP1 fluorescence and OD$_{600}$ were measured. Relative pP$_{cdrA}$::*mTFP1* reporter activity is the level of mTFP1 fluorescence normalized to OD$_{600}$. Asterisk indicates statistical significance by Student's t-test (p<0.001) in six technical replicates. Error bars = standard deviation. (**B**) The pP$_{cdrA}$::*mTFP1* reporter displays heterogeneity in a strain with a functional WspR (*wspR*[L170D]) and is consistently dark in a strain with inactive WspR. Histogram displaying the distribution of average cellular levels of mTFP1 fluorescence from expression of the pP$_{cdrA}$::*mTFP1* reporter in either the PAO1 Δ*wspR* attCTX::*wspR*[L170D]-eYFP (blue) or PAO1 Δ*wspR* attCTX::*wspR*[E253A]-eYFP (red) backgrounds. (**C**) Cells with subcellular clusters of functional WspR have higher levels of c-di-GMP reporter activity than cells without a WspR focus or cells without a functional WspR. L170D refers to a point mutant in WspR that forms large subcellular clusters and retains diguanylate cyclase activity: WspR[L170D]-eYFP in the strain PAO1 Δ*wspR* attCTX::*wspR*[L170D]-eYFP. E253A refers to a point mutant in WspR that forms large subcellular clusters but does not retain diguanylate cyclase activity: WspR[E253A]-eYFP in the strain PAO1 Δ*wspR* attCTX::*wspR*[E253A]-eYFP. Asterisk indicates statistical

*Figure 3 continued on next page*

*Figure 3 continued*

significance by Student's t-test (p<0.001) in four replicates, n.s. = not significant. Error bars = standard deviation. See *Figure 3—figure supplement 1* for a representative image of WspR[L170D]-eYFP foci.

The online version of this article includes the following source data and figure supplement(s) for figure 3:

**Source data 1.** *Figure 3B* source data.

**Figure supplement 1.** A representative image despicting subcellular foci of WspR[L170D]-eYFP in cells with high pP$_{cdrA}$::*mTFP1* reporter activity.

## Cyclic di-GMP heterogeneity leads to diversification in surface exploration at the lineage level

We hypothesized that heterogeneity in c-di-GMP signaling dictated by the Wsp complex could impact the surface behavior of the two observed subpopulations. We predicted that the subpopulation of cells with high c-di-GMP after surface attachment would produce biofilm matrix exopolysaccharides and contribute to initial microcolony formation, while the cells with low c-di-GMP would exhibit increased surface motility and detachment, which is known to be inhibited by exopolysaccharide production. To test this hypothesis, we tracked both reporter activity and surface behavior for cells within a single field of view for 40 hr. From our single-cell tracking data, we generated family trees across at least four generations of cells, using a previously described technique (*Lee et al., 2018*). We tracked the time-averaged P$_{cdrA}$::*gfp*ASV reporter activity (I$_{c-di-GMP}$), surface motility behavior (F$_{motile}$, defined as the fraction of time that cells are motile), and detachment behavior (tree asymmetry $\lambda$, where $\lambda = 0$ represents both daughter cells remaining attached to the surface and $\lambda = 1$ represents when one daughter cell detaches or travels outside the field of view).

In *P. aeruginosa*, surface exploration is mainly accomplished by type IV pili-mediated twitching motility, and does not appear to be influenced by levels of intracellular c-di-GMP when analyzing single cells (*Ribbe et al., 2017*). Interestingly, when analyzing correlations between c-di-GMP and motility for entire lineages in family trees, we found clear inverse correlations between I$_{c-di-GMP}$ and F$_{motile}$ (*Figure 4B*, $\rho = -0.53$, p=0.0012) and between I$_{c-di-GMP}$ and $\lambda$ (*Figure 4C*, $\rho = -0.45$, p=0.0068), suggesting that c-di-GMP levels are strongly inversely correlated with surface motility behavior and detachment behavior over multiple generation of cells. Analyzing these correlations across multiple cell divisions is important because it tells us when and how bacteria respond to c-di-GMP signaling events in terms of their surface motility and detachment. When looking at correlations between c-di-GMP and surface motility, the time between seeing a signaling event (i.e., rise or drop in reporter fluorescence intensity) and seeing a response (i.e., change in motility) can span a broad range (from a few minutes to well over a cell division time). Using a cell's entire lineage history (i.e., tracking daughter cells across cell divisions) will capture all of these events, whereas using single cell history (i.e., within one generation) will only capture a portion of them. For example, this lineage-level tracking of the influence of signaling events on bacterial behavior was recently shown for correlations between cyclic AMP signaling and *P. aeruginosa* surface motility (*Lee et al., 2018*). In this study, Lee *et al.* found that correlations between cell motility and signaling activity were stronger when they took into account lineage history, rather than using single cell history. However, if there are enough instances where the time between a signaling event and a cell's corresponding behavioral response are within a single generation, then correlations can still be found when using single cell history. For wild type PAO1 (WT), we observe weaker correlations when looking at individual cells in these lineages. I$_{c-di-GMP}$ vs F$_{motile}$ for single cells had a Spearman correlation value $\rho = -0.40$ (p<0.0001), which is smaller than the lineage-level correlation value, suggesting that lineage-level correlations are stronger.

To illustrate these correlations, we chose three representative families, with either high, intermediate, or low I$_{c-di-GMP}$ and plotted their family trees (*Figure 4D*) and spatial trajectories (*Figure 4E*). Families with the highest I$_{c-di-GMP}$ had the lowest F$_{motile}$ and $\lambda$ (Family 1, *Figure 4B–E*). In these families, daughter cells remained attached following cell division, exhibited continuously elevated c-di-GMP, did not move appreciable distances on the surface, and ultimately produced small microcolonies. In contrast, families of cells with low I$_{c-di-GMP}$ had the highest F$_{motile}$ and $\lambda$. For these families, daughter cells frequently detached or traveled outside the field of view, had lower c-di-GMP levels, traveled larger distances on the surface, and ultimately did not form microcolonies (Family 3, *Figure 4B–E*).

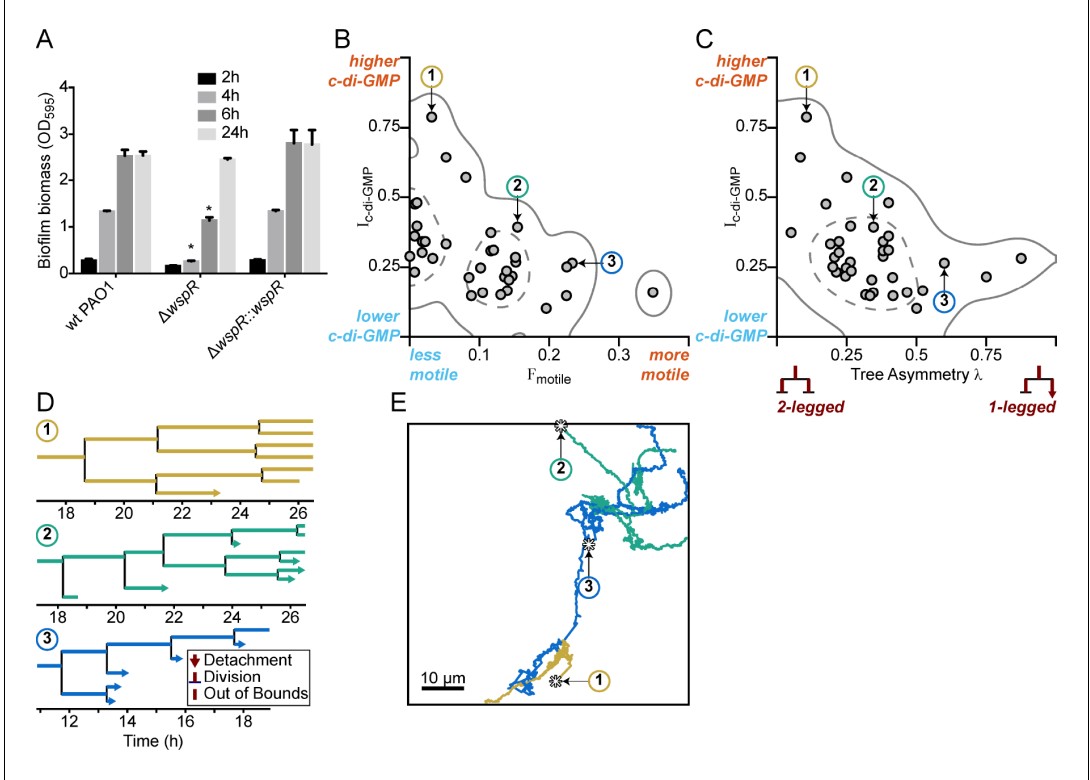

**Figure 4.** Multigenerational c-di-GMP levels within populations of surface-attached wild type PAO1 cells are inversely correlated with surface motility and detachment. (**A**) The Wsp surface sensing system is involved in the early stages of biofilm formation in PAO1. Static biofilm assay performed in wild type PAO1, a single deletion mutant of *wspR*, and the PAO1 Δ*wspR* mutant complemented with w*spR*. Between 4 and 6 hr, PAO1 Δ*wspR* shows a defect in surface attachment and biofilm formation relative to the wild type. However, after 24 hr, PAO1 Δ*wspR* formed equal biofilm biomass compared to wild type. Plotted values are the mean of 6 technical replicates and error is standard deviation. Asterisk indicates a statistically significant change in biomass relative to wild type PAO1 at each time point (Student's t test; p<0.05). (**B**) Plot of $I_{c-di-GMP}$ vs $F_{motile}$ for individual wild type PAO1 families. $I_{c-di-GMP}$ is the relative normalized c-di-GMP reporter intensity averaged across all members of a family. $F_{motile}$ is the fraction of time that cells in a family are motile (specifically surface translational motility). Each circle represents an individual family (N = 35) with at least four tracked generations. Solid lines represent the 95% probability bounds and dashed lines represent the 50% probability bounds, calculated via kernel density estimation. Spearman correlation: ρ = −0.53, p=0.0012. (**C**) Plot of $I_{c-di-GMP}$ vs tree asymmetry λ for individual wild type PAO1 families. Colored numbers indicate the same three families from (**B**) and (**D**). Tree asymmetry λ quantifies the detachment behavior of family trees as follows. λ = 0 corresponds to ideal trees with purely 'two-legged' division-branching, when both daughter cells remain attached to the surface. λ = 1 corresponds to ideal trees with purely 'one-legged' division-branching when one daughter cell detaches or travels outside the field of view. Points here are the same families as in (**B**). Solid lines represent the 95% probability bounds and dashed lines represent the 50% probability bounds, calculated via kernel density estimation. Spearman correlation: ρ = −0.45, p=0.0068. (**D**) Family trees of the same three representative wild type PAO1 families indicated in (**B**) and (**C**). Time 0 hr is the start of the dataset recording. Lengths of horizontal lines on the plots are proportional to time spent in each generation. Horizontal lines that end with arrows are detachment events, lines that intersect with a vertical line are division events, and lines that end without a marker are out-of-bound events where we lose track of the bacterium (moving out of the field of view or reaching the end of the recording; represented as moving outside the XYT limits of the dataset boundaries). Vertical lines are arbitrarily spaced to show all the descendants. Colors represent the families in (**B**) and (**C**). (**E**) Spatial trajectories of the three representative families. Asterisks (*) represent the initial location of the founder cell. Scale bar 10 µm. The families are color coded as in the previous panels.

The online version of this article includes the following source data and figure supplement(s) for figure 4:

**Source data 1.** MATLAB data for WT and Δ*wspR* families.

**Figure supplement 1.** The ΔwspR mutant does not have correlations between c-di-GMP, surface motility, and detachment.

Tracking and lineage analyses were also performed on a PAO1 mutant with the Wsp system inactivated (PAO1 Δ*wspR*). We observed that the range of tree asymmetry values is like that of WT and that the Δ*wspR* mutant eventually reaches WT c-di-GMP levels despite initially being lower, which is consistent with the observation that the mutant eventually forms WT-like biofilms. What our analysis revealed about the Δ*wspR* strain was quite unexpected: The WspR mutant has overall lower surface motility and, importantly, lacks correlations between c-di-GMP, surface motility, and detachment

behavior (both at the level of lineages and at the level of single cells). For single cell surface motility, 23% of $\Delta wspR$ mutant cells (48 of 210 cells) have non-zero $F_{motile}$ (the metric for surface motility) compared to 44% of WT cells (251 of 565 cells; *Figure 4—figure supplement 1*). The overall lowered surface motility and the lack of correlations between c-di-GMP and surface motility in the $\Delta wspR$ mutant suggest that the Wsp system is involved in translating the heterogeneous c-di-GMP signaling events into the corresponding motility-related responses for cells and their progeny. Therefore, the Wsp system's multi-generational temporal propagation of surface sensing signaling and behavior is important for the generation of heterogeneous populations of surface motile and immotile cells during early biofilm formation.

## Use of an optogenetic reporter to control bacterial surface behavior at the single-cell level

One important question is what happens to early biofilm development if we were to effectively remove heterogeneity in c-di-GMP output rooted in the WspR surface sensing system. To address this question, we used a strain in which c-di-GMP production could be easily controlled using an optogenetic system. The precise control of c-di-GMP expression in individual cells was made possible by the use of a chimeric protein that fused a diguanylate cyclase domain to a bacteriophytochrome domain. Flow chambers were seeded with the optogenetic strain encoding a heme oxygenase (*bphO*) and light-responsive diguanylate cyclase (*bphS*) (*Ryu and Gomelsky, 2014*). We initially characterized the optogentic strain and verified the reporter activity increased with exposure of the optogentic strain to red light (*Figure 5—figure supplement 1*) and that the laser light did not impact growth or motility (data not shown). Following validation of the strain, cells inoculated on a glass surface were tracked and continuously stimulated with red-light over ~8 hr using adaptive tracking illumination microscopy (ATIM), which allows for precise stimulation of the initial attached cells and their offspring and ensures sustained intracellular c-di-GMP production for a fixed number of surface cell generations (*Figure 5—figure supplement 2*). Cellular lineages (a cell and all of its offspring) and c-di-GMP reporter activity were continually monitored for at least 12 hr. Families that were not stimulated with light demonstrated a heterogeneous surface response (*Video 1* and *Figure 5B,D*) similar to that of Families 1–3 in *Figure 4B–E*. Some lineages were dominated by surface explorers, whereas others were seen to commit to microcolony formation. In contrast, in families stimulated with light for more than one generation, the resulting c-di-GMP production artificially forced lineages to have low surface motility and commit to microcolony production (*Video 1* and *Figure 5A,C*) similar to that of Family one in *Figure 4B–E*. Families stimulated with light in this manner had higher $I_{c\text{-di-GMP}}$ and lower $\lambda$ values than those that were not stimulated (*Figure 5—figure supplement 3*). We also found that optogenetic control of c-di-GMP results in phenotypes that are consistent with the wild-type behavior presented in *Figure 4*, with illuminated cells (high c-di-GMP) displaying the least motility and control (non-illuminated) cells displaying comparatively greater surface motility (*Figure 5—figure supplement 3*). Interestingly, families stimulated with light for one generation or less are not significantly different from un-illuminated controls (data not shown). Our data show that the generation of c-di-GMP can deterministically lead to the creation of an entire lineage of sessile cells with post-division surface persistence, low motility, and initiation of microcolony formation. Altogether, these results show that c-di-GMP levels, surface motility, and detachment are inversely correlated at the lineage level, and that the time scale for this occurs over multiple generations.

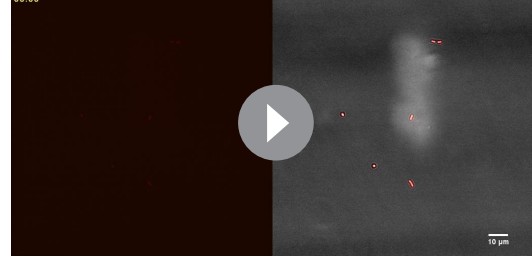

**Video 1.** Single cells are precisely illuminated by ATIM via *in situ* analysis and tracking of bacteria. The left panel shows the merged images of *gfp*ASV and mCherry fluorescence microscopy images over time. The right panel shows the merged images of red LED projected patterns and bright field images corresponding to the left panel. The fluorescence intensity of *gfp*ASV in the illuminated cells and their offspring (colored red in right panel) is significantly increased after using ATI for 460 mins. In contrast, the *gfp*ASV fluorescence intensity of the un-illuminated cells remains low and these cells remain motile.
https://elifesciences.org/articles/45084#video1

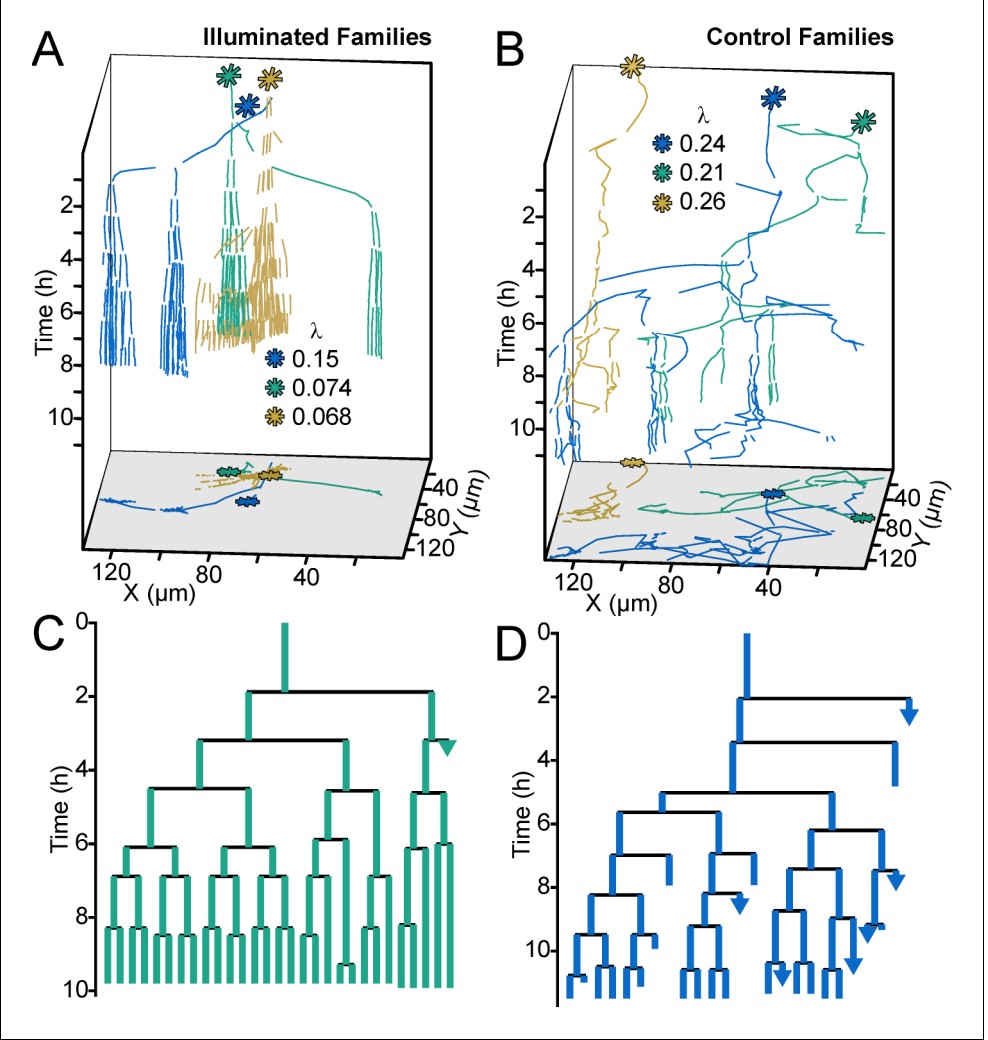

**Figure 5.** Optogenetic control of c-di-GMP production drastically affects family architecture and surface motility. (A,B) Spatiotemporal plot of 3 illuminated families (A) and three control families (B). The individual cell tracks in the 3D plot are projected onto the XY plane as spatial trajectories. As in *Figure 4*, λ is a measure of tree asymmetry, with higher values indicating more cells traveling outside the field of view or detaching. In A, the illuminated families tend to be sessile, as expected for cells with high c-di-GMP. In B, control cells are more motile than the illuminated cells in A. (C,D) Family trees of a single corresponding family in (A) and (B), where the color corresponds to the same family. Illuminated cells (C) tend to stay adhered across multiple generations, whereas control cells (D) display more surface motility and detachments. See *Video 1* for a representative video of the optogenetic reporter experiment. *Figure 5—figure supplement 1* is a control experiment showing that c-di-GMP levels increase in response to red light intensity in the optogenetic reporter strain. *Figure 5—figure supplement 2* shows a schematic of the ATIM apparatus. See *Figure 5—figure supplement 3* for the data from *Figure 5* overlayed onto *Figure 4C*, showing that ptogenetic-controlled families follow the trend of family behavior observed in wt PAO1 cells.

The online version of this article includes the following source data and figure supplement(s) for figure 5:

**Source data 1.** MATLAB data for optogenetic and WT families.

**Figure supplement 1.** Activity of the c-di-GMP reporter of PAO1-*bphS*-P$_{cdrA}$-GFP-mCherry was dependent on red-light intensity.

**Figure supplement 2.** Using Adaptive Tracking Illumination Microscopy (ATIM) to exactly illuminate single *P. aeruginosa* cells on surface.

**Figure supplement 3.** Optogenetic-controlled families follow the trend of family behavior observed in wt PAO1 cells, with illuminated families resembling the high c-di-GMP matrix producers and control families resembling low c-di-GMP surface explorers.

## Discussion

Collectively, our data show that heterogeneity in cellular levels of c-di-GMP generated by the Wsp system in response to surface sensing, leads to two physiologically distinct subpopulations that each contribute to surface colonization. Phenotypic heterogeneity of single cells is a common phenomenon in bacteria that can be beneficial at the population level by allowing a single genotype to survive sudden environmental changes. Sources of phenotypic heterogeneity among genetically homogeneous populations include bistability (*Dubnau and Losick, 2006*) and stochasticity (*Elowitz et al., 2002*) of gene expression, unequal partitioning of proteins during cell division due to low abundance (*Elowitz et al., 2002*), epigenetic modifications resulting in phase variation (*Casadesús and Low, 2006*), or through asymmetrical cell division (*Kulasekara et al., 2013*; *Laventie et al., 2019*). In this study, we show that the Wsp system generates heterogeneity in c-di-GMP signaling, and it is never fully activated in 100% of wild-type, surface-attached cells. One possible outcome of phenotypic heterogeneity is a division of labor between costly behaviors that support the growth and survival of the population (*Ackermann, 2015*). We found that abolishing c-di-GMP heterogeneity through inactivation of WspR leads to defects in early biofilm formation. This suggests that the subpopulations of high c-di-GMP, polysaccharide producers and low c-di-GMP, surface explorers are both required for efficient biofilm formation and that they represent a division of labor during early biofilm formation.

The data show that Wsp-generated c-di-GMP heterogeneity results in phenotypic changes for entire family lineages of cells. It is interesting that correlations between c-di-GMP, surface motility, and surface detachment probability are stronger when considered for an entire lineage in a bacterial family tree, but weaker when considered at the individual cell level. We think that this difference in the strength of correlations between c-di-GMP and bacterial behavior at the lineage versus single cell level is biologically meaningful and likely reflects the complex relationship between c-di-GMP signaling and type IV pili-mediated motility (*Ribbe et al., 2017*; *Jain et al., 2012*; *Jain et al., 2017*). The time between initiation of a signaling event (i.e., increased intracellular c-di-GMP) and the associated response (i.e., attenuation of motility or initiation of polysaccharide production) can span a large range, depending on the behavior. When examining polysaccharide production, signal propagation appears to be quick, with high c-di-GMP cells initiating exopolysaccharide production within minutes of *P. aeruginosa* encountering a surface. However, when examining a different bacterial behavior—surface motility—we found the strongest correlation between c-di-GMP signaling and cellular behavior at the lineage level. For many cells, there was a lag between when c-di-GMP first increased and surface motility decreased. While a mother cell may still be surface motile upon initiating c-di-GMP signaling, we found that following cell division, daughter cells with high c-di-GMP eventually became immotile. Thus, whereas an increase in c-di-GMP relatively quickly results in increased biofilm matrix production, this same increase in c-di-GMP likely indirectly influences surface motility. Supporting this, *P. aeruginosa* is known to produce Psl polysaccharide while engaging in type IV pili mediated motility across a surface, leaving behind a trail of Psl (*Zhao et al., 2013*). Finally, this apparent multigenerational influence of c-di-GMP signaling on bacterial behavior resembles the recently observed multigenerational memory of cAMP signaling in *P. aeruginosa* (*Lee et al., 2018*). Additional work is needed to determine whether surface-naïve daughter cells have any 'memory' of surface attachment by a mother cell, through the maintenance of elevated c-di-GMP across one or more cell divisions. Another future direction of this work is to examine whether other bacterial signaling modalities, including other nucleotide or non-nucleotide signaling systems (e.g. ppGpp or quorum sensing), may exhibit similar multigenerational features.

We observed that the Pil-Chp surface sensing system did not play a role in early c-di-GMP signaling in our experimental system. There are numerous potential explanations for this. For example, it is possible that the Pil-Chp system is the dominant surface sensing mechanism under different environmental conditions than the ones we used for this study. Alternatively, the Pil-Chp system might not contribute to general cytoplasmic pools of c-di-GMP (for which the reporter is sensitive) and instead participates in specific localized c-di-GMP signaling events. In support of this notion, inactivation of individual diguanylate cyclases is well known to lead to distinct changes in c-di-GMP-regulated behaviors (*Kulasakara et al., 2006*; *Merritt et al., 2010*). It is interesting to note that whereas the other Pil-Chp inactivation mutants tested did not have a phenotype, the *pilA* deletion mutant strain displayed a slight defect in surface sensing in our study, although why is currently unknown.

If we overwhelm WspR-generated c-di-GMP heterogeneity by using optogentically-induced sustained c-di-GMP production, we find that phenotypic heterogeneity is lost, and that illuminated cells deterministically become sessile and form microcolonies. Interestingly, our optogenetic experiments show that sustained c-di-GMP production for more than one generation is required before commitment to the sessile lifestyle. This observation is consistent with the fact that we see stronger correlations between c-di-GMP levels and motility behavior at the lineage level compared to the individual cell level. Moreover, since the Wsp surface sensing system generates heterogeneous c-di-GMP levels, this requirement of sustained c-di-GMP production for more than one generation is inherently difficult for wild-type cells to meet, and virtually guarantees the simultaneous existence of motile and sessile subpopulations.

The heterogeneity we observed in Wsp signaling shares many similarities with phenotypic heterogeneity generated from other c-di-GMP signaling (*Petersen et al., 2019*) and quorum sensing systems (*Grote et al., 2015*; *Ramalho et al., 2016*). Nucleotide second messenger and quorum sensing (QS) signaling systems are traditionally thought to coordinate cellular behavior in response to information regarding the cell's environment. However, rather than functioning to initiate a completely homogeneous response at the population level to environmental conditions, a growing body of literature suggests that a common theme of these signaling systems is that they introduce some level of behavioral heterogeneity (*Grote et al., 2015*). For example, QS-induced phenotypic heterogeneity in *Vibrio harveyi* is attributable to variability in the phosphorylation state of LuxO and influences bioluminencence and biofilm formation (*Anetzberger et al., 2009*). In *P. aeruginosa* and *Caulobacter crescentus*, heterogeneity in the very low levels of c-di-GMP present during planktonic growth is achieved through asymmetrical cell division and influences swimming motility (*Kulasekara et al., 2013*). Thus, *P. aeruginosa* appears to have at least two distinct mechanisms of generating c-di-GMP heterogeneity, which it employs during different modes of growth. In the case of the Wsp system, this phenotypic heterogeneity, which has been 'hardwired' into the structure of the Wsp surface sensing network, allows for a division of the labor during early biofilm formation, with one subpopulation committing to initiating the protective biofilm lifestyle, while the other subpopulation is free to explore the surface and potentially colonize distant, perhaps more favorable, locations.

# Materials and methods

**Key resources table**

| Reagent type (species) or resource | Designation | Source or reference | Identifiers | Additional information |
|---|---|---|---|---|
| Strain, strain background (*Pseudomonas aeruginosa*) | PAO1 | PMID: 111024 | | The version of PAO1 used in this study can be obtained from Matthew Parsek's laboratory. |
| Recombinant DNA reagent | pP*cdrA*::*gfp*ASV | PMID: 22582064 | | |
| Recombinant DNA reagent | PAO1 attCTX::*bphS* attMiniTn7::*mCherry* pP*cdrA*::*gfp*ASV | This study | NA | This strain can be obtained from Fan Jin's laboratory. |
| Commercial assay or kit | QuikChange Lightning Site-Directed Mutagenesis Kit | Agilent Technologies | 210519 | |
| Commercial assay or kit | Gateway BP Clonase II Enzyme mix | ThermoFisher Scientific | 11789020 | |
| Commercial assay or kit | LR Clonase II Plus enzyme | ThermoFisher Scientific | 12538120 | |
| Commercial assay or kit | QIAquick gel extraction kit | Qiagen | 28115 | |

*Continued on next page*

*Continued*

| Reagent type (species) or resource | Designation | Source or reference | Identifiers | Additional information |
|---|---|---|---|---|
| Commercial assay or kit | QIAquick PCR purification kit | Qiagen | 28104 | |
| Commercial assay or kit | Antarctic phosphatase | New England BioLabs | M0289S | |
| Commercial assay or kit | TRIzol LS | ThermoFisher Scientific | 10296010 | |
| Commercial assay or kit | RQ1 RNase-Free DNase | Promega | M6101 | |
| Commercial assay or kit | iTaq Universal SYBR Green One-Step kit | Bio-rad | 172–5150 | |
| Chemical compound, drug | TRITC Conjugated Wisteria floribunda lectin | EY laboratories | R-3101–1 | |
| Chemical compound, drug | TRITC Conjugated Hippeastrum hybrid Lectin (Amaryllis) | EY laboratories | R-8008–1 | |
| Software, algorithm | Volocity Image Analysis Software | Quorum Technologies Inc | Version 6.00 | |
| Software, algorithm | NIS-Elements AR | Nikon | Version 4.00 | |
| Software, algorithm | MATLAB code for tracking experiments | This study | Version R2015a | The code and datasets are available to download as source data files associated with *Figures 1*, *4* and *5*. |
| Software, algorithm | Statistics and Machine Learning Toolbox for MATLAB | MathWorks | Version R2015a | |
| Other | Flow cell for time course experiments | University of Iowa Machine Shop | Standard dimension flow cell | Flow cell dimensions: 5 mm x 35 mm x 1 mm |
| Other | Flow cell for cell tracking experiments | PMID: 18770573 | Standard dimension flow cell | This flow cell can be ordered from Department of Systems Biology, Technical University of Denmark. |

## Bacterial strains and growth conditions

The strains, plasmids, and primers used in this study are listed in *Table 1*. *Escherichia coli* and *P. aeruginosa* strains were routinely grown in Luria–Bertani (LB) medium and on LB agar at 37°C. For the flow cell experiments, *P. aeruginosa* was grown in either LB or FAB minimal medium supplemented with 10 mM or 0.6 mM glutamate at room temperature (*Zhao et al., 2013*). For flow cytometry experiments, *P. aeruginosa* was grown in either LB medium or in Jensen's defined medium with glucose as the carbon source (a growth medium in which Pel is more abundantly produced than in LB) (*Jennings et al., 2015*). For the tube biofilm and c-di-GMP measurements, *P. aeruginosa* strains were grown in Vogel-Bonner Minimal Medium (VBMM; *Vogel and Bonner, 1956*). Antibiotics were supplied where necessary at the following concentrations: for *E. coli*, 100 μg/mL ampicillin, 10 μg/mL gentamicin, and 10 or 60 μg/mL tetracycline; for *P. aeruginosa*, 300 μg/mL carbenicillin, 100 μg/mL gentamicin, and 100 μg/mL tetracycline. $P_{cdrA}$::$gfp_{ASV}$ reporter and vector control plasmids were selected with 100 μg/mL gentamicin for *P. aeruginosa* strains and 10 μg/mL gentamicin for *E. coli*.

PAO1 Δ*pilY1* was constructed using two-step allelic exchange following conjugation of wild type PAO1 with *E. coli* S17.1 harboring pENTRPEX18Gm::Δ*pilY1* (a gift from Joe Harrison) as previously described (*Hmelo et al., 2015*). PAO1 Δ*pilY1* was identified by colony PCR using primers PAO1pilY1-SEQ-F and PAO1pilY1-SEQ-R. PAO1 Δ*dipA* was constructed similarly by conjugation of wild type PAO1 with *E. coli* S17.1 harboring pENTRPEX18Gm::Δ*dipA* (a gift from Joe Harrison). PAO1 Δ*dipA* was identified by colony PCR using primers PAO1dipA-SEQ-F and PAO1dipA-SEQ-R. PA14

**Table 1.** Strains, primers, and plasmids used in this study.

| | | Reference |
|---|---|---|
| *P. aeruginosa* strains | | |
| PAO1 | wild-type | *Holloway et al., 1979* |
| PA14 | wild-type | *Rahme et al., 1997* |
| PAO1Δ*wspF* | markerless, in frame deletion of WspF | *Hickman et al., 2005* |
| PAO1Δ*wspF*Δ*pelA*Δ*pslBCD* | markerless, in frame deletions of WspF, PelA, and PslBCD genes | *Rybtke et al., 2012* |
| PAO1Δ*wspR* | markerless, in frame deletion of WspR | Hickman, 2005 |
| PAO1Δ*pilY1* | markerless, in frame deletion of PilY1 | this study |
| PAO1Δ*sadC* | markerless, in frame deletion of SadC | *Irie et al., 2012* |
| PAO1Δ*pilA* | markerless, in frame deletion of PilA | *Shrout et al., 2006* |
| PAO1Δ*dipA* | markerless, in frame deletion of DipA | this study |
| PAO1Δ*wspR* attCTX::P*wspA*::*wspR* | PAO1ΔwspR complemented with WspR under control of the Wsp operon promoter and including intergenic region upstream of WspR | Gift from Yasuhiko Irie |
| PAO1Δ*sadC* attCTX::*sadC* | PAO1ΔsadC complemented with SadC under control of its native promoter | Gift from Yasuhiko Irie |
| MPAO1 attTn7::P(A1/04/03)::GFPmut | wild type MPAO1 constitutively expressive stable GFP | this sudy |
| PA14 Δ*wspF* | markerless, in frame deletion of WspF | Gift from Caroline Harwood |
| PA14 Δ*wspR* | markerless, in frame deletion of WspR | Gift from Caroline Harwood |
| PAO1Δ*wspR* attCTX::PBAD-wspR-eYFP | markerless, in frame deletion of WspR with arabinose-inducible, C-terminally eYFP-tagged wild type WspR allele | *Huangyutitham et al., 2013* |
| PAO1Δ*wspR* attCTX::PBAD-wspR[L170D]-eYFP | markerless, in frame deletion of WspR with arabinose-inducible, C-terminally eYFP-tagged WspR[L170D] allele | *Huangyutitham et al., 2013* |
| PAO1Δ*wspR* attCTX::PBAD-wspR[E253A]-eYFP | markerless, in frame deletion of WspR with arabinose-inducible, C-terminally eYFP-tagged WspR[E253A] allele | *Huangyutitham et al., 2013* |
| *P. aeruginosa* reporter strains | | |
| PAO1 pMH489 | | *Rybtke et al., 2012* |
| PAO1 pP*cdrA*::*gfp*ASV | | *Rybtke et al., 2012* |
| PAO1 pP*siaA*::*gfp*ASV | | this study |
| PA14 pMH489 | | this study |
| PA14 pP*cdrA*::*gfp*ASV | | this study |
| PAO1Δ*wspF* pMH489 | | this study |
| PAO1Δ*wspF* pP*cdrA*::*gfp*ASV | | this study |
| PAO1Δ*wspF* pP*siaA*::*gfp* | | this study |

*Table 1 continued on next page*

Table 1 continued

| | | Reference |
|---|---|---|
| PAO1ΔwspFΔpelCΔpslD pMH489 | | this study |
| PAO1ΔwspFΔpelCΔpslD pPcdrA::gfpASV | | this study |
| PAO1ΔwspR pMH489 | | this study |
| PAO1ΔwspR pPcdrA::gfpASV | | this study |
| PAO1ΔwspR pPsiaA::gfp | | this study |
| PAO1ΔpilY1 pMH489 | | this study |
| PAO1ΔpilY1 pPcdrA::gfpASV | | this study |
| PAO1ΔsadC pMH489 | | this study |
| PAO1ΔsadC pPcdrA::gfpASV | | this study |
| PAO1ΔpilA pMH489 | | this study |
| PAO1ΔpilA pPcdrA::gfpASV | | this study |
| PAO1ΔdipA pMH489 | | this study |
| PAO1ΔdipA pPcdrA::gfpASV | | this study |
| PAO1ΔwspR attCTX:: PwspA::wspR pMH489 | | this study |
| PAO1ΔwspR attCTX::PwspA:: wspR pPcdrA::gfpASV | | this study |
| PAO1ΔsadC att::sadC pMH489 | | this study |
| PAO1ΔsadC att::sadC pPcdrA::gfpASV | | this study |
| PA14 ΔwspF pMH489 | | this study |
| PA14 ΔwspF pPcdrA::gfpASV | | this study |
| PA14 ΔwspR pMH489 | | this study |
| PA14 ΔwspR pPcdrA::gfpASV | | this study |
| PAO1ΔwspR attCTX:: PBAD-wspR-eYFP pPcdrA::mTFP1 | | this study |
| PAO1ΔwspR attCTX:: PBAD-wspR[L170D]-eYFP pPcdrA::mTFP1 | | this study |
| PAO1ΔwspR attCTX:: PBAD-wspR[E253A]-eYFP pPcdrA::mTFP1 | | this study |
| PAO1 attCTX:: bphS attMiniTn7::mCherry pPcdrA::gfp$_{ASV}$ | | this study |
| PAO1 attMiniTn7:: mCherry pPcdrA::gfp$_{ASV}$ | | this study |
| *E. coli* strains | | |
| E. coli S17.1 pENTRPEX18Gm::ΔpilY1 | conjugation proficient E. coli harboring pilY1 deletion allele | Gift from Joe Harrison |
| E. coli S17.1 pENTRPEX18Gm::ΔdipA | conjugation proficient E. coli harboring dipA deletion allele | Gift from Joe Harrison |
| E. coli DH5α pUC18-miniTn7T2-PcdrA-RBSg10L-gfpAGA | source of PcdrA-RBSg10L | this study |
| E. coli DH5α pBBR1MCS5-PcdrA::RBSg10L::mTFP1 | referred to as 'pPcdrA::mTFP1' | this study |
| E. coli DH5α pPsiaA::gfp | plasmid-based, fluorescent siaA transcriptional reporter | this study |
| Primers | | |

Table 1 continued

| | | Reference |
|---|---|---|
| PAO1pilY1-SEQ-F | CTACTACGAGACCAATAGCGTC | this study |
| PAO1pilY1-SEQ-R | GTCGATGTCCACCAGGTTCTTC | this study |
| PAO1dipA-SEQ-F | GATACGCTTAACTTGGGCCCTG | this study |
| PAO1dipA-SEQ-R | CTTTTCTTGGTGAGGATTTCAGAAC | this study |
| PA14wspR-SEQ-F | GCTTCCTCACCATCGCCC | this study |
| PA14wspR-SEQ-R | CAGGTCGTCCAGGGTTTCC | this study |
| PA14wspF-SEQ-F | CTCACGGTGCGTGAGCTG | this study |
| PA14wspF-SEQ-R | GGTCCTGGAGGATCACCG | this study |
| SacI – PcdrA - F | GGGGAGCTC GTATGGAA GGTTCCTTGGCGG | this study |
| SOE-PcdrA-RBSg10L - R | ctcctcgcccttgctcaccat GGATATATCTCCTTCTTAAAG | this study |
| mTFP1 - F | atggtgagcaagggcgaggag | this study |
| KpnI - mTFP1 – R | GGGGTACC ttacttgtacagctcgtcc | this study |
| BamH1-Psia-F | GGG GGATCC GGCAGCGGCAACCGCCTCTG | this study |
| SiaA-BamH1-R | CCC GGATCC CAACCCCCAGTTCGCCGCCAT | this study |
| M13F(−21) | TGTAAAACGACGGCCAGT | GeneWiz |
| M13R | CAGGAAACAGCTATGAC | GeneWiz |
| ampR-F-qPCR | GCG CCA TCC CTT CAT CG | *Colvin et al., 2011* |
| ampR-R-qPCR | GAT GTC GAC GCG GTT GTT G | *Colvin et al., 2011* |
| pslA-F-qPCR | AAG ATC AAG AAA CGC GTG GAA T | *Colvin et al., 2011* |
| pslA-R-qPCR | TGT AGA GGT CGA ACC ACA CCG | *Colvin et al., 2011* |
| pelA-F-qPCR | CCT TCA GCC ATC CGT TCT TCT | *Colvin et al., 2011* |
| pelA-R-qPCR | TCG CGT ACG AAG TCG ACC TT | *Colvin et al., 2011* |
| rplU-F-qPCR | CGC AGT GAT TGT TAC CGG TG | *Colvin et al., 2011* |
| rplU-R-qPCR | AGG CCT GAA TGC CGG TGA TC | *Colvin et al., 2011* |
| OBT268 | GGGGACAACTTTTGTATACAAAG TTGTACTATAGAGGGACAAACTC AAGGTCATTCGCAAGAGTGGC CTTTATGATTGACCTTCTTCCGG TTAATACGACCGGGATAACTCCAC TTGAGACGTGAAAAAAGAGGAGTA TTCATGCGTAAAGGAGAAGAA CTTTTCACTGGAG | This study |
| OBT269 | GGGGACAAGTTTGTACAAAAAAGCA GGCTCGGCTTATTTGTATAGTTCAT CCATGCCATGTGTAATC | This study |
| OBT314 | CAGGTCGACTCTAGAGGATCCCCATC AGAAAATTTATCAAAAAGAGTGTTGACT TGTGAGCGGATAACAATGATACTTAGATT CAATTGTGAGCGGATAACAATTTCACA CATCTAGAATTAAAGAGGAGAAATTAA GCATGGTGAGCAAGGGCGAGGAG | *Zhao et al., 2013* |
| OBT315 | CTCCTCGCCCTTGCTCACCATGCTTAA TTTCTCCTCTTTAATTCTAGATGTGT GAAATTGTTATCCGCTCACAATTGAA TCTAAGTATCATTGTTATCCGC TCACAAGTCAACACTCTTTTT GATAAATTTTCTGATGGGGAT CCTCTAGAGTCGACCTG | *Zhao et al., 2013* |

Table 1 continued on next page

Table 1 continued

| | | Reference |
|---|---|---|
| pP*cdrA::gfp*ASV | P*cdrA* reporter with short halflife GFP | *Rybtke et al., 2012* |
| pENTRPEX18Gm::Δ*pilY1* | suicide plasmid containing *pilY1* deletion construct for use in PAO1 | Gift from Joe Harrison |
| pENTRPEX18Gm::Δ*dipA* | suicide plasmid containing *dipA* deletion construct for use in PAO1 | Gift from Joe Harrison |
| pBBR1MCS5 | broad host range vector that is stable in *P. aeruginosa*, GentR | *Elzer et al., 1995* |
| pUC18-miniTn7T2-P*cdrA*-RBSg10L-*gfp*AGA | source plasmid containing promoter of *cdrA* with enhanced ribosomal binding site | this study |
| pNCS-mTFP1 | source plasmid containing mTFP1 | Allele Biotech |
| pBBR1MCS5-PcdrA::RBSg10L::mTFP1 | teal fluorescent protein version of P*cdrA* reporter | this study |
| pP*siaA::gfp* | P*siaA* reporter expressing stable GFP, constructed using pMH487 plasmid | this study |
| pBT270 | miniTn7 transposon with *gfpmut3* driven by the A1/04/03 promoter; Apr, Gmr | This study |
| pTNS2 | T7 transposase expression vector | *Choi and Schweizer, 2006* |
| pBT223 | miniTn7 transposon with *gfpmut3* driven by the *trc* promoter; Apr, Gmr | This study |
| pBT212 | A GateWay compatible plasmid containing *gfpmut3* flanked by attR5 and attL1 recombination sites; Kmr | This study |
| pBT200 | A GateWay compatible plasmid containing the *trc* promoter flanked by attL2 and attL5 recombination sites; Knr | *Zhao et al., 2013* |
| pUC18-miniTn7T2-Gm-GW | A GateWay compatible mini-Tn7 based vector; Cmr, Apr and Gmr; | *Zhao et al., 2013* |
| AKN66 | source for *gfpmut3* | *Lambertsen et al., 2004* |
| pDONR221 P1-P5r | A GateWay compatible vector with attP1 and attP5r recombination sites and *ccdB*; Knr and Cmr | Invitrogen |

Δ*wspR* and Δ*wspF* deletion mutants were confirmed by PCR using primers PA14wspR-SEQ-F and PA14wspR-SEQ-R or PA14wspF-SEQ-F and PA14wspF-SEQ-R, respectively.

To create MPAO1 attTn7::P(A1/04/03)::GFPmut, the miniTn7 from pBT270 was integrated into the chromosome of *P. aeruginosa* PAO1 with the helper plasmid pTNS2, as previously described (*Choi and Schweizer, 2006*). pBT270 was created by introducing the constitutive A1/04/03 promoter (*Lanzer and Bujard, 1988*) and removing the trc promoter from pBT223 using the Quik-Change Lightning Kit (Agilent Technologies) and the oligonucleotides OBT314 and OBT315. pBT223 was constructed via recombineering of pBT200, pUC18-miniTn7T2-Gm-GW, and pBT212 using

Multisite Gateway technology (Invitrogen). pBT212 was constructed by cloning the *gfpmut3* from AKN66 using OBT268 and OBT269, and recombining the PCR product with pDONR221 P1-P5r.

## Construction of optogenetic, c-di-GMP reporter strain in *P. aeruginosa*

Chromosomal insertion of *bphS* was achieved using the mini-CTX system and these strains were marked with different fluorescent proteins by mini-Tn7 site-specific transposition essentially as previously described (*Choi and Schweizer, 2006*; *Hoang et al., 2000*). First, a *bphS* fragment obtained from the plasmid pIND4 was cloned into the vector mini-CTX2 with the *PA1/O4/O3* promoter upstream of the MCS via a two-piece ligation. The constructed plasmid was electroporated into PAO1 and the corresponding recombinant strain was identified by screening on LB agar plates containing 1 mM IPTG and 100 µg/mL tetracycline. Then, the strains were electroporated with a pFLP2 plasmid and distinguished on LB agar plates containing 5% (w/v) sucrose for the excision of the resistance marker. The c-di-GMP reporter plasmid and mCherry/EGFP marked *bphS* mutants were constructed as described above. The c-di-GMP reporter plasmid ($P_{cdrA}::gfp_{ASV}$) was electroporated into the mCherry-marked strain harboring *bphS* to monitor the intracellular c-di-GMP level.

To validate the optogenetic reporter strain in *Figure 5—figure supplement 1*, strains were grown on LB agar plates at 37°C for 24 hr from frozen stocks. Monoclonal colonies were inoculated and cultured with a minimal medium (FAB) at 37°C overnight, adding 1 uM $FeCl_3$ and 30 mM glutamate as the carbon source, until the culture reached an $OD_{600}$ of approximately 2.1. Then, the bacterial culture was further diluted (1:100) in fresh FAB medium to $OD_{600}$ 0.5. When required, gentamicin was added to medium at 30 µg/mL. Plates and tubes were wrapped with aluminum foil to achieve a dark condition. Finally, the culture was diluted (1:50) in fresh FAB medium and 6 µL diluted culture was spotted onto an FAB agarose (2%) pad, with 30 mM glutamate and 1 µM $FeCl_3$. The agarose pad was pressed on a coverglass before cells were illuminated.

## Cyclic di-GMP measurement and qRT-PCR of tube biofilms

Measurement of c-di-GMP in tube biofilm cells was performed as previously described (*Colvin et al., 2011*). Transcriptional analysis of PelA expression in tube biofilms was performed as described in the 'FACS and qRT-PCR of c-di-GMP reporter cells' section.

## Crystal violet attachment assays

Crystal violet assays were performed essentially as previously described to measure biofilm biomass, except using gentle washing after 2–6 hr of static incubation (*Armbruster et al., 2016*). To measure biofilm biomass at 24 hr, the crystal violet assay was performed as previously described without gentle washing (*Colvin et al., 2012*).

## Flow cell time course experiments and confocal microscopy

*P. aeruginosa* cells harboring the $pP_{cdrA}::gfp_{ASV}$ reporter plasmid or a promotorless vector control (pMH489) were grown to mid-log in LB with 100 µg/mL gentamicin (Gm100) from LB Gm100 plates or from overnight broth cultures in FAB +10 mM glutamate. Mid-log cells were back diluted into 1% LB or FAB +0.6 mM glutamate and flow chambers were inoculated at a final $OD_{600}$ 0.1 and inverted for 10 min to allow cells to attach before induction of flow. Clean media was used to wash non-attached cells by flow at 40 mL per hour for 20 min. Flow was then reduced to a final constant flow rate of 3 mL per hour and bacteria were imaged immediately on a Zeiss LSM 510 scanning confocal laser microscope (t = 0 hr). Flow cells were incubated at a constant flow rate at room temperature and imaged hourly for up to 24 hr. For every strain and time point, 5 fields of view and a minimum of 300 cells were captured using identical microscope settings to image GFP fluorescence across all experiments. Images were analyzed using using Volocity software (Improvision, Coventry, UK). We binned cells by their mean GFP fluorescence intensity per pixel, in incremints of 20 fluorescence units, and determined the cut-off bin that corresponded to cells clearling produced GFP when images were examined by eye. Therefore, cells were counted as $pP_{cdrA}::gfp_{ASV}$ reporter 'on' if their mean GFP fluorescence intensity per pixel was $\geq$321 fluorescence units. For all summary figures depicting the percentage of cells with the reporter 'on', data are presented in terms of the percentage of cells with an average GFP fluorescence per pixel of $\geq$321 fluorescence units ($pP_{cdrA}::gfp_{ASV}$ reporter 'on'). Microscopy images were artificially colored to display GFP fluorescence as green.

## Construction of pP*siaA*::*gfp*

A region 259 bp upstream through 21 bp into the coding sequence of *siaA* was amplified from PAO1 genomic DNA using primers BamH1-Psia-F and SiaA-BamH1-R, then gel purified using a QIA-quick gel extraction kit (Qiagen, Hilden, Germany) digested with BamH1, then column purified with a QIAquick PCR purification kit (Qiagen, Hilden, Germany) to remove BamH1. The GFP expression vector pMH487, which contains the *gfp*mut3 gene with an RNase III splice site and lacking a promoter (*Borlee et al., 2010*), was digested with BamH1, treated with Antarctic phosphatase (New England Biolabs, Ipswich, MA), then column purified with a QIAquick PCR purification kit (Qiagen, Hilden, Germany) to remove BamH1. The P*siaA* allele was ligated into digested pMH487, then transformed into *E. coli* DH5α, purified, and sequenced using primer M13F(−21) (Genewiz). The reporter pP$_{siaA}$::*gfp* was electroporated into *P. aeruginosa* as previously described and maintained under gentamycin selection at 100 µg/mL.

## Multi-generation single cell tracking of type IV motility and c-di-GMP reporter activity

Wild type PAO1 harboring the pP$_{cdrA}$::*gfp*$_{ASV}$ reporter was grown shaking for 20 hr in FAB media with 6 mM glutamate. The flow cell inoculum was prepared by diluting the culture to a final OD$_{600}$ of 0.01 in FAB with 0.6 mM glutamate. The flow cell inoculum was injected into the flow cell (Department of Systems Biology, Technical University of Denmark) and allowed to incubate for 10 min at 30°C prior to flushing with media at 30 mL/h for 10 min. Experiments were performed under a flow rate of 3 mL/hour for a total of 40 hr.

Images were acquired with an Olympus IX81 microscope equipped with a Zero Drift Correction autofocus system, a 100 × oil objective with a 2 × multipler lens, and an Andor iXon EMCCD camera using Andor IQ software. Bright-field images were recorded every 3 s and GFP fluorescence every 15 min. Acquisition continued for a total recording time of 40 hr, which resulted in approximately 48000 bright-field images, and 160 fluorescence images.

Images were analyzed in MATLAB to track bacterial family trees, GFP fluorescence, and surface motility essentially as previously described (*Lee et al., 2018*) with the following modifications. Image analysis, family tracking and manual validation, family tree plotting, and tree asymmetry λ calculations were performed as previously described (*Lee et al., 2018*) without modification. GFP fluorescence intensities were normalized by calculating the distribution of intensities per cell per frame (extracted by using the binary image as a mask) and then setting the minimum and maximum intensities to the 1$^{st}$ and 99$^{th}$ percentiles of this distribution for each dataset. I$_{c-di-GMP}$ (relative normalized c-di-GMP reporter intensity) was calculated by averaging the normalized fluorescence intensities across all members of a family. F$_{motile}$ (fraction of time that cells in a family are motile) was calculated as follows. For each family, every cell trajectory in the family was divided into time intervals. For each time interval, presence or absence of motility was determined using a combination of metrics, including Mean Squared Displacement (MSD) slope, radius of gyration, and visit map. MSD slope quantifies the directionality of movement relative to diffusion. Radius of gyration and visit map are different metrics for quantifying the average distance traveled on the surface. F$_{motile}$ was then calculated by the fraction of these time intervals that have motility. This calculation was modified from the 'TFP activity metric' previously described (*Lee et al., 2018*).

## Setup of Adaptive Tracking Illumination Microscopy

*Figure 5—figure supplement 2* shows a schematic of the Adaptive Tracking Illumination Microscopy (ATIM) setup. An inverted fluorescent microscope (Olympus, IX71) was modified to build the ATIM. The modification includes: 1) a commercial DMD-based LED projector (Gimi Z3) was used to replace the original bright-field light source, in which the original lenses in the projector were removed and three-colored (RGB) LEDs were rewired to connect to an external LED driver (ThorLabs) controlled by a single chip microcomputer (Arduino UNO r3); 2) the original bright-field condenser was replaced with an air objective (40× NA = 0.6, Leica); and 3) an additional 850 nm LED light (Thor-Labs) was coupled to the illumination optical path using a dichroic mirror (Semrock) for the bright-field illumination. The 850 nm LED bright field light source does not affect optogenetic manipulation. An inverted fluorescent microscope (Olympus, IX71) equipped with a 100× oil objective and a sCMOS camera (Zyla 4.2 Andor) was used to collect bright-field images with 0.2 frame rate. The

bright-field images were further analyzed to track multiple single cells in real time using a high-throughput bacterial tracking algorithm coded by Matlab. The projected contours of selected single cells were sent to the DMD (1280 × 760 pixels) that was directly controlled by a commercial desktop through a VGA port. The manipulation lights were generated by the red-color LED (640 nm), and were projected on the single selected cells in real time through the DMD, a multi-band pass filter (446/532/646, Semrock) and the air objective. Our results indicated that feedback illuminations could generate projected patterns to exactly follow the cell movement (Figure 5 – Supplemental 2B) or single cells divisions (Figure 5 – Supplemental 2C) in real time.

## Manipulation of c-di-GMP expression in single initial-attached cells

The bacterial strain PAO1-*bphS*-P*cdrA*-GFP-mCherry was inoculated into a flow cell (Denmark Technical University) and continuously cultured at 30.0 ± 0.1 ˚C by flowing FAB medium (3.0 mL/h). The flow cell was modified by punching a hole with a 5 mm diameter into the channel, and the hole was sealed by a coverslip that allows the manipulation light to pass through. An inverted fluorescent microscope (Olympus, IX71) equipped with a 100× oil objective and a sCMOS camera (Zyla 4.2 Andor) was used to collect bright field or fluorescent images with 0.2 or 1/1800 frame rate respectively. The power density of the manipulation lights was determined by measuring the power at the outlet of the air objective using a power meter (Newport 842-PE). GFP or mCherry was excited using a 480 nm or 565 nm LED lights (ThorLabs) and imaged using single-band emission filters (Semrock): GFP (520/28 nm) or mCherry (631/36 nm). Initial-attached cells were selected to be manipulated using ATIM with the illumination at 0.05 mW/cm$^2$, which allowed us to compare the results arising from illuminated or un-illuminated mobile cells in one experiment. The c-di-GMP levels in single cells were gauged using the ratio of GFP and mCherry intensities.

## Lectin staining and flow cytometry

Glass culture tubes were inoculated with 1 mL of *P. aeruginosa* in LB or Jensen's minimal media at an OD$_{600}$ 0.8 and incubated statically at 37˚C for 4 hr. Non-adhered cells were removed by washing three times with 2 mL sterile phosphate buffered saline (PBS). Biofilm cells were harvested by vortexing in 1 mL PBS with tetramethylrhodamine (TRITC) conjugated lectins (TRITC-labeled WFL lectin (100 μg/mL; Vector Laboratories) for Pel, TRITC-labeled HHA (100 μg/mL; EY Laboratories) for Psl) and incubated on ice for 5 min. Cells were washed 3 times to remove non-adhered lectin, resuspended in PBS, and immediately analyzed for GFP and TRITC fluorescence on a BD LSRII flow cytometer (BD Biosciences). Events were gated based on forward and side scatter to remove particles smaller than a single *P. aeruginosa* cell and large aggregates.

We used PAO1 cells that did not express GFP (wild type PAO1; *Figure 1—source data 2*) or constitutively expressed GFP (PAO1 Tn7::P(A1/04/03)::GFPmut; *Figure 1—source data 2*) to define a gate for high GFP fluorescence. We validated this gate using a strain in which we expect very high levels of reporter activity (surface grown PAO1 Δ*wspF*Δ*pelA*Δ*pslBCD* harboring pP*cdrA::gfp*$_{ASV}$) and saw that 91.6% of cells had high GFP levels (*Figure 1—source data 2*), in agreement with our flow cell characterization of this strain (*Figure 2A*). We determined gating for TRITC using cells that had not been stained with TRITC-conjugated lectin (*Figure 1—figure supplement 6A*), as well as two strains that overproduced either Psl (*Figure 1—figure supplement 6B*) or Pel (*Figure 1—figure supplement 6C*) that were stained with the appropriate TRITC-conjugated lectin. Our flow cytometry gating procedure accurately gated 99.7% of wild type PAO1 cells (without the P*cdrA* reporter or lectin-staining) as low GFP and low TRITC (*Figure 1—figure supplement 6D*).

## FACS and qRT-PCR of c-di-GMP reporter cells

Static biofilm reporter cells were grown as described above and harvested without lectin staining. Cells were fixed with 6% paraformaldehyde for 20 min on ice, then rinsed once with sterile PBS prior to analysis with a FACSAriaII (BD Biosciences, San Jose, CA). Events were gated first to remove debris and large cellular aggregates, and then gated into cells with low and high GFP fluorescence intensity. The low GFP gate was drawn using wild type PAO1 cells without the gfp gene (*Figure 1—figure supplement 7A*) and the high GFP gate was drawn using both PAO1 Tn7::P(A1/04/03)::GFPmut (*Figure 1—figure supplement 7B*) and PAO1 Δ*wspF* Δ*pelA* Δ*pslBCD* P$_{cdrA}$::*gfp*$_{ASV}$ reporter (*Figure 1—figure supplement 7C*). As expected, wild type PAO1 pP$_{cdrA}$::*gfp*$_{ASV}$ reporter cells that

had been harvested after 4 hr of surface attachment to glass in static LB liquid culture displayed sub-populations of high GFP, reporter 'on' cells (30.8% of the population) and 'off' (57.2%) cells (*Figure 1—figure supplement 7D*), whereas this same strain grown to mid-log planktonically in LB displayed mostly reporter 'off' cells (*Figure 1—figure supplement 7E*). Cells were sorted at 4°C by flow assisted cell sorting (FACS) to collect 100,000 events into TRIzol LS (Thermo Fisher Scientific, Waltham, MA). RNA was extracted from sorted cells by boiling immediately for 10 min and following the manufacturer's instructions for RNA isolation. DNA was digested by treating with RQ1 Dnase I (Promega, Madison, WI) and samples were checked for genomic DNA contamination by PCR to detect *rplU*. Expression of *pelA*, *pslA*, and *ampR* was measured by quantitative Reverse Transcriptase PCR (qRT-PCR) using the iTaq Universal SYBR Green One-Step kit (Biorad, Hercules, CA) and a CFX96 Touch Real-Time PCR detection system (Bio-Rad, Hercules, CA). The $\Delta\Delta C_q$ was calculated for three independent samples of sorted wild type PAO1 $P_{cdrA}::gfp_{ASV}$ reporter biofilm cell populations by normalizing PelA and PslA to relative levels of AmpR expression. Data were presented as the average fold change in PelA or PslA expression in the $P_{cdrA}::gfp_{ASV}$ sorted 'on' population (high GFP) relative to the 'off' population (low GFP) for the three biological replicates.

### WspR-YFP foci and pP*~cdrA~*::*mTFP1* reporter

A version of the pP*cdrA* reporter was constructed in the pBBR1MCS5 plasmid to express mTFP1 instead of GFP, for use with YFP-tagged WspR proteins. The P*cdrA* promoter and an enhanced ribosomal binding site from the gene 10 leader sequence of the T7 phage (g10L) was amplified from pUC18-miniTn7T2-P*~cdrA~*-RBSg10L-*gfp*~AGA~ using primers SacI-PcdrA-F and SOE-PcdrA-RBSg10L-R. The primers mTFP1-F and KpnI-mTFP1-R were used to amplify the mTFP1 gene from plasmid pNCS-mTFP1 (Allele Biotech, San Diego, CA). The P*~cdrA~*::RBSg10L::*mTFP1* allele was constructed by SOE-PCR using primers SacI-PcdrA-F and Kpn1-mTFP1-R, then pBBR1MCS5 and the SOE PCR product were doubly digested with SacI/KpnI. Digested pBBR1MCS5 was treated with Antarctic phosphatase, then both digests were gel purified and ligated. The ligation was transformed into *E. coli* DH5α, and plasmid from clones growing on LB with 10 μg/mL gentamycin were sequenced with primers M13F and M13F(−21) (GeneWiz). Fluorescence of the pP*~cdrA~*::*mTFP1* reporter was measured in Wsp mutants in a fluorimeter (BioTek Synergy H1 Hybrid Reader, BioTek Instruments, Inc, Winooski, VT, USA) and in flow cells to confirm its activity resembled that of pP*~cdrA~*::*gfp*~ASV~. The pP*cdrA*::*mTFP1* reporter was electroporated into *P. aeruginosa* strains with the native WspR deleted and harboring an arabinose-inducible copy of WspR-YFP on its chromosome (*Huangyutitham et al., 2013*). Cells were grown on LB agar plates with 100 μg/mL gentamycin and 1% arabinose for 10 hr, then transferred to an agar pad for imaging of Differential Interference Contrast (DIC), YFP, and TFP. WspR-YFP foci and mTFP1 fluorescence was imaged using a Nikon Ti-E inverted wide-field fluorescence microscope with a large-format scientific complementary metal-oxide semiconductor camera (sCMOS; NEO, Andor Technology, Belfast, United Kingdom) and controlled by NIS-Elements. WspR-YFP foci were detected and pP*cdrA*::*mTFP1* reporter activity were analyzed using NIS-Elements AR software (Nikon Instruments, Melville, NY, USA). Regions of interest (ROI) corresponding to individual cells were determined using DIC images and the average mTFP1 fluorescence was measured within these ROIs. Next, we used essentially the same protocol as previously described for detecting WspR-eYFP foci (*Huangyutitham et al., 2013*), by examining the ratio of the maximum eYFP signal to the mean eYFP signal for each ROI. We verified by eye that the previously determined cut-off ratio of 1.7 accurately represented cells with at least one visible WspR-eYFP focus (*Huangyutitham et al., 2013*).

## Acknowledgements

We thank Drs. Julie Cass and Paul Wiggins providing the wide-field microscope and cMOS camera to image WspR-eYFP clusters, and Drs. Joe J Harrison and Yasuhiko Irie for the gift of bacterial strains.

# Additional information

## Funding

| Funder | Grant reference number | Author |
| --- | --- | --- |
| National Institutes of Health | T32GM007270 | Catherine R Armbruster |
| Charlie Moore Endowed Fellowship | | Catherine R Armbruster |
| Army Research Office | W911NF1810254 | Calvin K Lee<br>Gerard CL Wong<br>Matthew R Parsek |
| National Institutes of Health | K22AI121097 | Boo Shan Tseng |
| National Institute of General Medical Sciences | GM56665 | Caroline S Harwood |
| National Natural Science Foundation of China | 21474098 | Fan Jin |
| Fundamental Research Funds for the Central Universities | WK2340000066 | Fan Jin |
| National Institutes of Health | K24HL141669 | Lucas R Hoffman |
| National Institutes of Health | 5R01AI077628 | Matthew R Parsek |
| National Natural Science Foundation of China | 21774117 | Fan Jin |
| National Natural Science Foundation of China | 21522406 | Fan Jin |
| Fundamental Research Funds for the Central Universities | WK3450000003 | Fan Jin |
| National Institutes of Health | 1R01AI143730-01 | Calvin K Lee<br>Gerard CL Wong |
| National Institutes of Health | R01AI143916 | Matthew R Parsek |

The funders had no role in study design, data collection and interpretation, or the decision to submit the work for publication.

## Author contributions

Catherine R Armbruster, Conceptualization, Data curation, Formal analysis, Supervision, Funding acquisition, Investigation, Visualization, Methodology, Writing—original draft, Project administration, Writing—review and editing; Calvin K Lee, Conceptualization, Data curation, Formal analysis, Investigation, Visualization, Methodology, Writing—review and editing; Jessica Parker-Gilham, Data curation, Formal analysis, Investigation, Visualization; Jaime de Anda, Data curation, Formal analysis, Investigation, Methodology; Aiguo Xia, Data curation, Formal analysis, Visualization; Kun Zhao, Data curation, Investigation, Methodology; Keiji Murakami, Data curation; Boo Shan Tseng, Conceptualization, Supervision, Funding acquisition, Investigation, Methodology, Writing—review and editing; Lucas R Hoffman, Supervision, Funding acquisition, Methodology, Writing—review and editing; Fan Jin, Conceptualization, Resources, Formal analysis, Supervision, Funding acquisition, Investigation, Methodology, Writing—review and editing; Caroline S Harwood, Conceptualization, Resources, Funding acquisition, Methodology, Writing—review and editing; Gerard CL Wong, Conceptualization, Resources, Supervision, Funding acquisition, Methodology, Project administration, Writing—review and editing; Matthew R Parsek, Conceptualization, Supervision, Funding acquisition, Methodology, Project administration, Writing—review and editing

## Author ORCIDs

Catherine R Armbruster [ID] https://orcid.org/0000-0003-0795-802X
Calvin K Lee [ID] https://orcid.org/0000-0001-6789-0317
Boo Shan Tseng [ID] http://orcid.org/0000-0001-7563-0232

Fan Jin ⬤ https://orcid.org/0000-0003-2313-0388
Matthew R Parsek ⬤ https://orcid.org/0000-0003-2932-7966

**Decision letter and Author response**
Decision letter https://doi.org/10.7554/eLife.45084.sa1
Author response https://doi.org/10.7554/eLife.45084.sa2

## Additional files

### Supplementary files
• Source code 1. Source code for fluorescence statistical tests.

• Source code 2. Source code for family tree plots.

• Source code 3. Source code for optogenetic family tree plots.

• Transparent reporting form

### Data availability
Source data files and/or MATLAB code have been provided for Figures 1, 3, 4, and 5.

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
