## [Decision Letter]

Thank you for submitting your article "Heterogeneity in surface sensing produces a division of labor in *Pseudomonas aeruginosa* populations" for consideration by *eLife*. Your article has been reviewed by three peer reviewers, including Alexandre Persat as the Reviewing Editor and Reviewer #1, and the evaluation has been overseen by Gisela Storz as the Senior Editor.

The reviewers have discussed the reviews with one another and the Reviewing Editor has drafted this decision to help you prepare a revised submission.

Summary:

This manuscript describes how heterogeneity in cyclic-di-GMP among *P. aeruginosa* cells drives the early stages of biofilm formation. Specifically, the authors provide evidence that this variation arises largely through the response of the Wsp surface sensing system.

*P. aeruginosa* proceeds to form biofilms by initially exploring surfaces. During this stage, surface sensing systems are activated and respond by expressing genes required for robust biofilm formation. At early times, the Pil-Chp system promotes the expression of genes associated with high levels of the second messenger molecule cyclic AMP. Genes associated with the biofilm state are overexpressed through activation of multiple other two-component systems that regulate levels of c-di-GMP, another second messenger. Some of these systems are sensitive to surface contact or shear on longer timescales. However, how single cell surface sensing and increase in c-di-GMP at the single cell level lead to the formation of biofilms remains unknown. Here, the authors track single cell c-di-GMP levels in surface-associated *P. aeruginosa* populations to probe how heterogeneity in these levels leads to successful biofilm formation.

This manuscript is filling a gap in our understanding of how surface sensing and c-di-GMP signaling leads to biofilm formation.

Essential revisions:

We have a few suggestions that we think will help improve the manuscript for publication in *eLife*.

1) In Figures 1 and 2, measurements are based on a percentage of population being above a threshold fluorescence. The threshold is calculated from background value (not sure if this is cell or camera background) but nevertheless this is arbitrary. The authors should focus on highlighting the complete distribution of fluorescence, including some analysis on the evolution of fluorescence intensity over time for single cells.

2) Figure 2B assumes that a standing cell has been standing for the whole duration of the experiment, but c-di-GMP levels are likely a result of its history. Thus, information on history of these cells is missing. To draw a conclusion on standing vs. non-standing one needs to look at history of a given cell.

3) The data presented demonstrates heterogeneity in surface sensing. However, there is no clear demonstration of a link of heterogeneity with surface sensing, and even less of a division of labor. The title is therefore misleading and should probably be revised, unless the authors provide more evidence for the division of labor. A division of labor and phenotypic variation do not necessarily go hand in hand.

4) Do c-di-GMP levels vary among planktonic cells prior to binding the surface? The variance of the mean in c-di-GMP levels in planktonic cells (Figure 1A) likely represents technical noise, but the broad distribution of c-di-GMP reporter activities in the flow cytometry experiments suggests that the so-called planktonic cells that "leave" the biofilm due to low c-di-GMP (Figure 4) will have varying levels. Figure 1—figure supplement 2 shows that the reporter activity at time 0 (i.e. planktonic) appears "off" but this alone does not confirm that the c-di-GMP levels are not already varying below the detectable level. The main reason why we bring this point to the table is that the detectable heterogeneity in reporter activity, which is already evident at 1 hour (Figure 1—figure supplement 2), could be simply due to the amplification of pre-existing heterogeneity even before "surface sensing".

5) As the author showed that cells lacking WspR can still form biofilms after 24h (Figure 4A). It would be interesting to show whether there is a tree asymmetry or a difference in motility (like Figure 4B and C) in WspR deletion mutants. This would definitely show the importance of the Wsp system in generating the heterogeneity of phenotypes in the population upon surface contact. Tracking and lineage analysis should be done in the *wspR* deletion mutant.

6) Subsection “Cyclic di-GMP heterogeneity leads to diversification in surface exploration at the lineage level”, last paragraph: The optogenetic construct does not appear to have been validated in *P. aeruginosa*. Can the authors independently confirm that the construct increases c-di-GMP, either by measuring c-di-GMP or through lectin staining?

7) "…pP*_cdrA_* activity was significantly higher in cells with at least one subcellular WspR-eYFP focus…" It is very difficult for the reader to come to this conclusion based on the images given. Can the authors provide quantification of pP*_cdrA_* for no foci and with foci?

---

## [Author Response]

Essential revisions:We have a few suggestions that we think will help improve the manuscript for publication in eLife.1) In Figures 1 and 2, measurements are based on a percentage of population being above a threshold fluorescence. The threshold is calculated from background value (not sure if this is cell or camera background) but nevertheless this is arbitrary. The authors should focus on highlighting the complete distribution of fluorescence, including some analysis on the evolution of fluorescence intensity over time for single cells.

We recognize that showing summary data in which cells are classified as being above or below a threshold fluorescence value loses information on single cell’s fluorescence intensity. One possibility is that while the overall percentage of “on” cells remains constant, the mean/median fluorescence intensity of the “on” cells may change over time in a biologically meaningful way. To address this comment, we have performed new analyses of the fluorescence intensity of single cells of wild type PAO1 over 12 hours, resulting in two new figures to accompany Figure 1.

First, new Figure 1—figure supplement 3 shows individual fluorescence values of wild type PAO1 reporter cells over the course of 12 hours and a dotted line is drawn at the cut-off for “off” versus “on” used throughout this study. We also overlaid the median and interquartile range on each timepoint. Examining changes in the median fluorescence value allows us to investigate whether the long tails of GFP fluorescence particularly at 2 and 4 hours represent an early “spike” in c-di-GMP production that levels off over time. Consistent with the message in the initial submission, each timepoint after 0 hours, we can clearly see a subpopulation of “off” cells with GFP fluorescence intensities similar to that of the 0 hours and vector control and an “on” population that increases over time, but never reaches 100% of the population. These results are added in the first paragraph of the subsection “The P*_cdrA_::gfp* reporter suggests heterogeneity in c-di-GMP levels during surface sensing”.

We then examined the evolution of fluorescence intensity over time for single cells by bootstrap sampling of the single cell fluorescence intensity values in the new Figure 1—figure supplements 4 and 5 (p values of time points in Figure 1—figure supplement 4). The purpose of this bootstrapping analysis is to examine whether the distributions of fluorescence intensity differ at each time point. We found that the median fluorescence intensity was significantly different at every timepoint except between 4 and 6 hours, and between 8, 10, and 12 hours. This single cell analysis supports a model in which c-di-GMP signaling is initiated rapidly upon surface attachment in the first 4-6 hours, then the rate of c-di-GMP increase tends to level off at later timepoints. These results have been added to the second paragraph of the aforementioned subsection.

Finally, we also addressed the reviewers’ concern about the “on/off” cut-off value being arbitrary by clarifying how the cut-off is determined in the Materials and methods subsection “Flow cell time course experiments and confocal microscopy”. Briefly, we found that given the microscope settings used throughout this study, a cell with a mean fluorescence intensity of greater than 320 was visible by eye as expressing GFP. To validate this cutoff value of “on” versus “off”, we confirmed that our vector control strain (the same plasmid containing GFP, but lacking a promoter) was reliably classified as “off” across many experiments, as expected. Importantly, we used the same cut-off value of “on” versus “off” throughout all experiments and have clarified this point throughout the manuscript as needed.

2) Figure 2B assumes that a standing cell has been standing for the whole duration of the experiment, but c-di-GMP levels are likely a result of its history. Thus, information on history of these cells is missing. To draw a conclusion on standing vs. non-standing one needs to look at history of a given cell.

We acknowledge this important point and respectfully disagree with the reviewers’ suggestion that we assume standing cells have been standing for the entire experiment. These data are snapshots in time of cells that are either polarly or laterally-attached, with a certain c-di-GMP level. To address this comment, we have clarified in the Results that the distinction of polar versus lateral refers to the cell’s attachment state at that point in time (and does not reflect the cell’s history) (subsection “The Wsp system is required for surface sensing”, last paragraph).

We strongly agree with the reviewers on the importance of history, particularly in the history of the entire lineage and not just a single cell. Notably, the polar versus lateral designation of cells in this figure was heavily manually curated because we currently lack a reliable way to make this distinction, while taking into account the fact that growing and dividing cells may be misclassified. This type of analysis, which relies on the ratio of the length to width of the cell, is technically very challenging. For example, a shorter cell that is laterally attached can have the same apparent length/aspect ratio as a more tilted and longer cell. We are currently developing a better technique based on using finite element methods to calculate all diffracted light fields and then perform cross correlation calculations between measured and calculated images to reduce ambiguity in tilt state and utilizing this technique in longitudinal studies (de Anda et al., ACS Nano, 2017).

3) The data presented demonstrates heterogeneity in surface sensing. However, there is no clear demonstration of a link of heterogeneity with surface sensing, and even less of a division of labor. The title is therefore misleading and should probably be revised, unless the authors provide more evidence for the division of labor. A division of labor and phenotypic variation do not necessarily go hand in hand.

We agree that phenotypic variation and “division of labor” are not equivalent terms. To address this important point and to better support our claim that the subpopulations we identified represent a division of labor specifically, we have expanded our review of these topics as they pertain to biofilm formation and c-di-GMP in the Introduction and expanded our discussion of division of labor in the conclusion section. Specifically, we have linked the heterogeneity we have seen in the surface sensing response to the heterogeneity observed an already published in the Wsp system. Briefly, we found that the Wsp system is required for the phenotypic heterogeneity we observed and that inactivation of WspR leads to defects in early biofilm formation (Figure 4A). This defect in biofilm formation caused by the *wspR* mutation, which abolishes the specialization of cells into surface explorers and polysaccharide producers, supports our claim that this specialization is a division of labor. We have specifically addressed this point in the Results (subsection “Heterogeneity in c-di-GMP levels among cells correlates with Wsp system activity”, last paragraph) and Discussion (first paragraph). In addition, we have amended the title of the article.

4) Do c-di-GMP levels vary among planktonic cells prior to binding the surface? The variance of the mean in c-di-GMP levels in planktonic cells (Figure 1A) likely represents technical noise, but the broad distribution of c-di-GMP reporter activities in the flow cytometry experiments suggests that the so-called planktonic cells that "leave" the biofilm due to low c-di-GMP (Figure 4) will have varying levels. Figure 1—figure supplement 2 shows that the reporter activity at time 0 (i.e. planktonic) appears "off" but this alone does not confirm that the c-di-GMP levels are not already varying below the detectable level. The main reason why we bring this point to the table is that the detectable heterogeneity in reporter activity, which is already evident at 1 hour (Figure 1—figure supplement 2), could be simply due to the amplification of pre-existing heterogeneity even before "surface sensing".

We thank the reviewers for this thoughtful line of inquiry regarding whether c-di-GMP heterogeneity exists prior to surface sensing, which brought to our attention an important citation that we inadvertently left out of our original manuscript. We agree that the variance in c-di-GMP levels at t = 0 hours in our assays likely represents technical noise; however, *P. aeruginosa* cells are known to display heterogeneity at very low c-di-GMP levels. We have addressed this question through expanding the review of literature on c-di-GMP heterogeneity in the Introduction section (fifth paragraph), by including a new figure (Figure 2—figure supplement 5) and Results (subsection “The Wsp system is required for surface sensing”, first paragraph), and by including a discussion of c-di-GMP heterogeneity during planktonic versus biofilm growth (Discussion, last paragraph).

In Kulasekara et al. (2013), they used a FRET-based c-di-GMP reporter to show that planktonic *P. aeruginosa* has heterogeneous, albeit very low, concentrations of c-di-GMP which it achieves through asymmetrical partitioning of a phosphodiesterase (PA5017, also called Pch) to the flagellated cell pole during cell division. We have currently ruled out this mechanism as responsible for the heterogeneity we see during surface sensing by examining a mutant of PA5017 and showing that this strain still exhibited heterogeneity during surface sensing. We feel that the inclusion of this experiment strengthens our conclusion that the heterogeneity we observe during surface sensing represents a distinct mechanism.

The reviewers also bring up an interesting point relevant to a previous paper (Lee et al., 2018). Cells can attach to and detach and reattach to the surface during the early stages of biofilm formation that we are monitoring. Because of this, sampling the “planktonic” liquid population inside the flow cell at a given time point will give you a heterogeneous mixture of cells that may or may not have been surface-associated at some point. In principle, one can develop the technology to simultaneously track the full multigenerational history of c-di-GMP reporter activity for both cells on the surface and cells swimming in the flow cell volume over the course of biofilm formation. We are working along these directions, but they are outside the scope of the present paper.

5) As the author showed that cells lacking WspR can still form biofilms after 24h (Figure 4A). It would be interesting to show whether there is a tree asymmetry or a difference in motility (like Figure 4B and C) in WspR deletion mutants. This would definitely show the importance of the Wsp system in generating the heterogeneity of phenotypes in the population upon surface contact. Tracking and lineage analysis should be done in the wspR deletion mutant.

Tracking and lineage analysis have been performed on the Δ*wspR* mutant. From this analysis, we see that the range of tree asymmetry values is like that of WT and that the Δ*wspR* mutant eventually reaches WT c-di-GMP levels despite initially being lower, which is consistent with the observation that the mutant eventually forms WT-like biofilms. What our analysis revealed about Δ*wspR* is quite unexpected: The mutant has overall lower surface motility and importantly, unlike WT, lacks correlations between c-di-GMP, surface motility, and detachment behavior (both at the level of lineages and at the level of single cells). For single cell surface motility, 48/210=23% of Δ*wspR* mutant cells vs. 251/565=44% of WT cells have non-zero F_motile_, the metric for surface motility. The overall lowered surface motility and the lack of correlations between c-di-GMP and surface motility in the Δ*wspR* mutant suggest that the Wsp system is involved in translating the heterogeneous c-di-GMP signaling events into the corresponding motility-related responses for cells and their progeny. This multi-generational temporal propagation of surface sensing signaling and behavior is important for the generation of heterogeneous populations during biofilm formation. One important point we want to emphasize is that one cannot consider the temporal evolution of a single cell independently of the temporal evolution of that cell’s entire lineage. These new data are discussed in the last paragrapah of the subsection “Cyclic di-GMP heterogeneity leads to diversification in surface exploration at the lineage level”.

6) Subsection “Cyclic di-GMP heterogeneity leads to diversification in surface exploration at the lineage level”, last paragraph: The optogenetic construct does not appear to have been validated in P. aeruginosa. Can the authors independently confirm that the construct increases c-di-GMP, either by measuring c-di-GMP or through lectin staining?

To monitor the intracellular c-di-GMP levels, we constructed the c-di-GMP reporter plasmid and electroporated it into mCherry marked *bphS* mutant (see original manuscript Materials and methods). The c-di-GMP levels in single cells were gauged using the ratio of GFP and mCherry intensities (F_G_/F_R_, original manuscript). Huang et al. (2018) had described how they manipulated the c-di-GMP levels via the optogenetic tools in *P. aeruginosa*. To address this comment and validate the c-di-GMP levels dependence of the red light intensity, we have now measured the ratio of GFP and mCherry of strain PAO1-*bphS* -P*_cdrA_*-GFP-mCherry after illumination with red light of variable intensity(0~0.268 mW/cm^2^) and obtained the curve of the time versus the ratio of GFP and mCherry (new Figure 5—figure supplement 1), which had shown that the c-di-GMP levels had increased significantly when strains was illuminated at more than 0.038 mW/cm^2^. This new validation is described in the Results subsection “Use of an optogenetic reporter to control bacterial surface behavior at the single-cell level” and in new Figure 5—figure supplement 1.

7) "…pP_cdrA_ activity was significantly higher in cells with at least one subcellular WspR-eYFP focus…" It is very difficult for the reader to come to this conclusion based on the images given. Can the authors provide quantification of pP_cdrA_ for no foci and with foci?

We agree with the reviewers that the original Figure 3B needed a quantitative analysis. We have now quantified reporter activity in cells with and without a WspR focus. We replaced the old Figure 3C with this quantification, moved the microscopy image to Figure 3—figure supplement 1, and added information regarding this image analysis to the Materials and methods subsection “WspR-YFP foci and pP*_cdrA_::mTFP1* reporter”. Importantly, this quantification confirms that cells with a WspR focus have much higher reporter activity, supporting a role for the Wsp system in introducing c-di-GMP heterogeneity during surface sensing.